# BottleneckMLP: Graph Explanation via Implicit Information Bottleneck

## Abstract

The success of Graph Neural Networks (GNNs) in modeling unstructured data has heightened the demand for explainable AI (XAI) methods that provide transparent, interpretable rationales for their predictions. A prominent line of work leverages the Information Bottleneck (IB) principle, which frames explanation as optimizing for representations that maximize predictive information $I(Z; Y)$ while minimizing input dependence $I(X; Z)$. We show that explicit IB-based losses in GNN explainers provide little benefit beyond standard training: the fitting and compression phases of IB emerge naturally, whereas the variational bounds used in explicit objectives are too loose to meaningfully constrain mutual information. To address this, we propose BottleneckMLP, an architectural module that implicitly enforces the IB principle. By injecting Gaussian noise inversely scaled by node importance, followed by architectural compression, BottleneckMLP amplifies the reduction of $I(X; Z)$ while increasing $I(Z; Y)$. This yields embeddings where important nodes remain structured and clustered, while unimportant nodes drift toward Gaussianized, high-entropy distributions, consistent with progressive information loss under IB. BottleneckMLP integrates seamlessly with current explainers, as well as subgraph recognition tasks, replacing explicit IB terms and consistently improving predictive performance and explanation quality across diverse datasets.

## 1 Introduction

Graph-structured data appears in a wide range of domains, including drug design (Liu et al., 2023), healthcare (Zitnik et al., 2018), social networks (Bian et al., 2020), and recommendation systems (Chen et al., 2022). Graph Neural Networks (GNNs) have emerged as powerful models for learning from such data, achieving state-of-the-art results in tasks such as node/graph classification (Bacciu et al., 2019; Kipf & Welling, 2017), link prediction (Zhang & Chen, 2018), and graph regression (Zhang et al., 2024); yet, GNNs remain black-box models. Recent research has focused on developing explainability methods (Dai et al., 2024; Li et al., 2023b; Yuan et al., 2022), which are essential to build trust and ensure reliability in sensitive applications such as healthcare and scientific discovery. These methods aim to find a graph explanation as a subgraph that is necessary and sufficient to explain the downstream GNN task.

Most GNN explainability methods are post-hoc (Bajaj et al., 2021a; Baldassarre & Azizpour, 2019; Luo et al., 2020a), applying an explainer to a pre-trained black-box model. Recent work instead explores ante-hoc approaches (Luong et al., 2024; Miao et al., 2022; Seo et al., 2024a), which train the explainer and classifier jointly to avoid spurious correlations. These intrinsically interpretable GNNs aim to balance accuracy with interpretability, encouraging reliance on ground-truth explanatory features. We focus on ante-hoc graph classification explainers, and additionally show that BottleneckMLP generalizes to post-hoc node classification and subgraph recognition tasks.

The Information Bottleneck (IB) principle (Tishby et al., 2000; Tishby & Zaslavsky, 2015) formalizes the tradeoff: the optimal representation $Z$ should capture minimal but sufficient information from $X$ to predict $Y$. The IB principle is pertinent to graph data, where rich structure and feature dependencies make learning compact, task-relevant representations challenging. Previous work in GNN explainers (ante-hoc (Miao et al., 2022), post-hoc (Chen et al., 2024), prototype-based (Seo et al., 2024b)) include IB losses to encourage the learned representation to be sufficient yet minimal.

In this work, we argue that explicitly minimizing $I(X;Z)$ via auxiliary explicit IB losses ('IB Loss') is ineffective. We formalize GNN explanation as an IB problem, noting that cross-entropy training alone induces the information curve: deeper layers naturally compress $X$ (input graph $G$), retaining only the information in $Z$ (explanatory subgraph $G_S$) needed for predicting $Y$. We reconstruct the information plane for GNNs and show consistency with IB theory and empirical findings in representation learning (Tishby & Zaslavsky, 2015). Our method surpasses IB-based explainers without such terms, improving accuracy and explanation quality. Our key contributions are fourfold:

(1) **Ineffectiveness of Explicit IB in Graph Settings** We show that explicit IB loss terms are ineffective for explanation in the graph setting, as structural dependencies violate the i.i.d. assumptions underlying IB theory.

(2) **BottleneckMLP: A General Implicit Architectural IB Module** We propose Bottle-neckMLP, a model-agnostic architectural primitive that implicitly enforces the IB principle without variational bounds or auxiliary losses. By injecting importance-scaled Gaussian noise and applying architectural compression (via an MLP), it drives unimportant nodes toward Gaussianized high-entropy embeddings while preserving structured clusters of important nodes, yielding compact, interpretable representations.

(3) **Gaussianization Encourages Forgetting** We show that the Gaussianization effect induced by BottleneckMLP serves as a natural mechanism for forgetting task-irrelevant information. By injecting noise inversely proportional to node importance, uninformative node representations are progressively pushed toward high-entropy, Gaussian-like embeddings, removing spurious correlations. Relevant information is retained in clustered, low-entropy embeddings, while irrelevant information is forgotten through Gaussianization.

(4) **Empirical Validation** In comprehensive experiments on ante-hoc graph explainers, post-hoc node classification explainers, and subgraph recognition models, BottleneckMLP consistently outperforms explicit IB losses in both explanation quality and model performance, demonstrating that implicit IB is more effective for interpretable graph learning.

## 2 PRELIMINARIES

**Mutual information** (MI) is a symmetric measure of how much one random variable reduces the uncertainty of another random variable:

$$I(X;Z) = H(X) - H(X|Z), \tag{1}$$

where $H(X)$ denotes the entropy of random variable $X$. (A complete list of all symbols and notation used in the paper appears in Appendix A.)

**Information Bottleneck**, proposed in (Tishby et al., 2000), provides a framework to learn a minimal sufficient representation $Z$ that preserves only the aspects of $X$ relevant to $Y$, quantified by the mutual information $I(X;Y)$. Assuming $X$ and $Y$ are statistically dependent, with $Y$ implicitly distinguishing between relevant and irrelevant features in $X$, the goal is for $Z$ to retain all information needed to predict $Y$ while discarding irrelevant details. The general IB objective is:

$$\min_{p(z|x)} \left[ -I(Z;Y) + \beta I(X;Z) \right], \quad \beta \geq 0 \tag{2}$$

where the Lagrange multiplier $\beta$ balances relevant information $I(Z;Y)$ and compression $I(X;Z)$.

**IB in Deep Learning.** Tishby & Zaslavsky (2015) interpret DNN training via the IB framework, where Shwartz-Ziv & Tishby (2017) identify two distinct phases in MI dynamics:

(1) **Fitting Phase**: Model fits the data; minimizing cross-entropy increases $I(X;Z)$ and $I(Z;Y)$.

(2) **Compression Phase**: Layers discard task-irrelevant information, reducing $I(X;Z)$ while preserving or increasing $I(Z;Y)$. Proceeds slowly and without explicit regularization.

The two-phase dynamic of DNN training emerges in our graph experiments due to BottleneckMLP, marking the first such observation in the graph setting. In graph learning, this behavior is especially desirable as embeddings must encode rich structural, node, and edge-level features into compact, low-dimensional representations. We leverage this insight in our architectural mechanism that achieves stronger implicit compression while improving predictive accuracy and explanation quality.

## 3 RELATED WORK

### 3.1 IB-BASED EXPLAINERS

A key challenge in generating explanatory subgraphs is their varying size, making fixed-size constraints ineffective (Kakkad et al., 2023). To address this, these information-constrained methods adopt the IB principle (Tishby et al., 2000), which limits retained information rather than subgraph size. Ante-hoc explainers such as GSAT (Miao et al., 2022), PGIB (Seo et al., 2024b), and TGIB (Seo et al., 2024a) incorporate IB objectives to encourage minimal sufficient representations:

$$\min_{\phi} -I(G_S; Y) + \beta I(G_S; G), \quad \text{s.t. } G_S \sim g_\phi(G). \tag{3}$$

GSAT learns edge attention weights to suppress irrelevant features, sampling $G_S$ from $P_\phi(G_S|G)$. PGIB introduces prototypes $G_p$ and modifies the objective to include $I(Y; G_S, G_p)$ and $I(Y; G_p|G_S) + \beta I(G; G_S)$. TGIB, a temporal variant, extracts bottleneck subgraphs $R_k$ from temporal neighborhoods, using an analogous IB objective. While these methods rely on variational upper bounds or contrastive loss to constrain $I(X; Z)$, we show these bounds are too loose to enforce meaningful compression. In contrast, our BottleneckMLP achieves the compression phases of the IB curve effectively, without requiring explicit IB loss terms.

### 3.2 COMPRESSION IN DEEP LEARNING

Several studies have shown that DNNs naturally undergo an implicit compression phase during supervised training. Scabini & Bruno (2023) use complex network theory to show that emergent motifs arise during training without explicit regularization, supporting the IB perspective (Tishby & Zaslavsky, 2015). Similarly, simple fully connected layers improve CNN generalization (Basha et al., 2020; Kocsis et al., 2022) even without explicit compression losses. These results suggest that architectural biases alone can induce compact, informative representations. Our BottleneckMLP demonstrates this in GNN explainability; compression can emerge directly from architecture and cross-entropy training without IB loss terms, bridging implicit IB dynamics with graph explanations.

### 3.3 GAUSSIANIZATION OF REPRESENTATIONS

(Eftekhari & Papyan, 2025) show that Gaussian distributions are both the most efficient signal representation and the worst-case noise. They propose a mechanism that enforces Gaussianity in neural representations, where injecting Gaussian noise and normalization improve generalization and robustness across architectures. Agrawal et al. (2020) extend infinite-width theory to bottleneck neural network Gaussian processes (NNGPs), showing that unlike deep ReLU NNGPs which lose discriminative power, bottleneck layers preserve task-relevant information by acting as information-preserving compression points. These results suggest that structured architectures (e.g., bottlenecks, Gaussianized activations) naturally promote robustness and generalization, supporting our view that compression and explainability need not rely on explicit IB constraints. Our BottleneckMLP follows this principle, achieving effective compression and explanation purely through architectural design.

## 4 BOTTLENECKMLP

We begin by identifying critical limitations of explicit IB-based approaches in graph explainability (Section 4.1), then introduce our BottleneckMLP module as an architectural solution (Section 4.2). We provide theory for why this approach is more effective than explicit IB losses (Section 4.3), followed by empirical validation of our theoretical results (Sections 4.3.3 and 5).

### 4.1 LIMITATIONS OF EXPLICIT INFORMATION BOTTLENECK IN GRAPH EXPLAINABILITY

While some post-hoc explainers (Bajaj et al., 2021b; Luo et al., 2020b) impose sparsity, budget, or connectivity constraints on explanatory subgraphs, the predominant strategy across IB-based methods remains the use of variational upper bounds. However, these approaches face fundamental limitations when applied to graph-structured data. We describe the general approach used by these ante-hoc IB explainers below, and give explainer-specific details in Appendix I.

**Variational Upper Bounds are Loose** The terms of the Lagrangian in Equation 2 cannot be computed directly, and require the integrals for $I(X; Z)$ and $I(Z; Y)$. The marginal $p(z)$ and the true posterior $p(y|z)$ are intractable, and variational bounds for mutual information (Poole et al., 2019) are used in machine learning (Du et al., 2020; Dai et al., 2018; Bao, 2021; Li et al., 2023a). Maximizing $I(Z; Y)$ reduces to the usual cross-entropy (CE) loss:

$$I(Z; Y) = \mathbb{E}_{p(y,z)}\left[\log p(y|z)\right] - H(Y). \tag{4}$$

Almost all ante-hoc graph explainers (Lee et al., 2023; Miao et al., 2022; 2023; Seo et al., 2024a;b) rely on variational approximations or naive priors which lead to loose or ineffective bounds. This results in insufficient enforcement of the information constraint, allowing the learned subgraphs to retain excessive or redundant information from the input. The variational upper bound loss $\mathcal{L}_{VUB}$ for graph explainers is defined as:

$$\mathcal{L}_{VUB} := \mathrm{KL}(P_\phi(G_S \mid G) \,\|\, Q(G_S)), \tag{5}$$

where edges $e_{ij}$ in $Q(G_S)$ are parameterized by $e_{ij} \sim \mathrm{Bernoulli}(r)$ edges $\hat{e}_{ij}$ are parameterized by $\hat{e}_{ij} \sim Bernoulli(\phi_{ij})$. We refer to this as simply 'IB Loss' throughout the paper. PGIB uses a contrastive loss (Appendix I to minimize $I(X; Z)$, which we again refer to as 'IB Loss'.

**Structural Dependencies Break Node-Level Independence Assumptions** GNNs compute node embeddings via recursive message passing: $h_v^{(k)} = \mathtt{UPDATE}^{(k)}(h_v^{(k-1)}, \mathtt{AGGREGATE}^{(k)}(\{h_u^{(k-1)} : u \in N(v)\}))$. Since each $h_v^{(k)}$ aggregates information from multi-hop neighborhoods, node features are interdependent. This violates the i.i.d assumptions underlying variational IB bounds, making KL-based regularizers over node distributions ineffective in capturing structural dependencies. We elaborate on this in Section 4.3.1.

**Empirical Evidence of Ineffectiveness** Sections 4.3.3 and 5 provide empirical results showing that explicit IB constraints fail to regulate information effectively, motivating our implicit, architecture-driven alternative based on selective forgetting of information.

## 4.2 BOTTLENECKMLP ARCHITECTURE

We propose BottleneckMLP, a two-component architectural module that implicitly enforces the IB principle without requiring explicit IB Loss terms.

**Component 1: Importance-Weighted Gaussian Noise** For each node embedding $Z_i \in \mathbb{R}^d$ with importance scores $\alpha_i \in (0, 1)$, we inject noise as:

$$Z_i = f(X) + \varepsilon_i, \quad \varepsilon_i \sim \mathcal{N}\left(0, (\frac{\sigma^2}{\alpha_i})I_d\right), \tag{6}$$

where $\sigma$ is a tunable hyperparameter, $f(X)$ is the GNN embedder, and $I_d$ is the $d$-dimensional identity matrix. This mechanism scales the variance of injected Gaussian noise inversely with importance, so that nodes deemed unimportant are perturbed more heavily, thereby pushing them toward high-entropy representations. We present our theoretical results for the component in Section 4.3.2

**Component 2: Progressive Compression via MLP Layers** Following Tishby & Zaslavsky (2015), we use an MLP to compress representations and filter task-irrelevant noise. The default configuration is $h \to \frac{h}{4} \to h$, with ReLU activations between layers. Choosing the optimal BottleneckMLP architecture for each dataset-explainer pair is analogous to hyperparameter tuning. This compression retains salient information while discarding noise, promoting compact, meaningful representations.

**Integration with Existing Explainers** BottleneckMLP integrates seamlessly into existing ante-hoc explanation pipelines (Figure 1), operating directly on the GNN embeddings before subgraph extraction. This modular design replaces explicit IB loss terms across different explainer architectures. The expansion layer is not essential for inducing bottleneck behavior; it primarily ensures architectural modularity so that BottleneckMLP can be inserted into existing models without altering embedding dimensionality. This mirrors prior IB literature, showing that any dimensionality reduction introduces a bottleneck, regardless of whether the dimension is restored afterward (Tishby et al., 2000). Thus, while the compression stage drives the information bottleneck, the expansion simply preserves compatibility with downstream architectures.

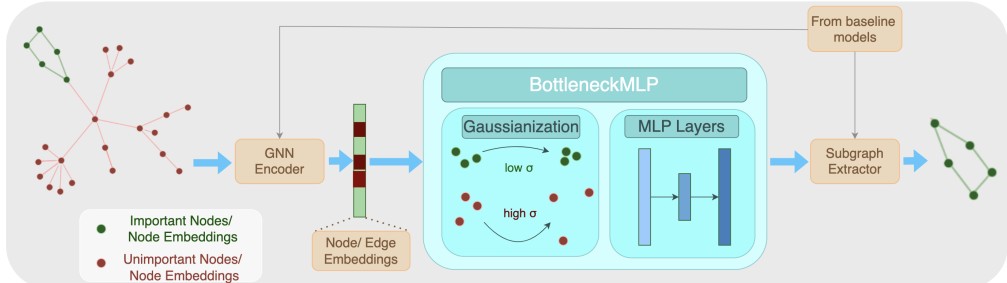

Figure 1: BottleneckMLP method for a general ante-hoc (intrinsically interpretable) GNN pipeline. Our general module acts on the embeddings produced by the explainer's GNN module, and the transformed embeddings are directed toRU the subsequent subgraph extractor component.

**Importance Score Computation** Node importance scores are computed using the explainer's existing attention or selection mechanism. GSAT uses attention weights from the stochastic attention mechanism, PGIB uses prototype similarity scores, and TGIB uses temporal neighborhood relevance scores. This ensures compatibility while leveraging each method's inherent importance estimation.

## 4.3 THEORY

In the following subsections, we provide theoretical justification for why explicit IB methods fail on graphs (Section 4.3.1), and grounding for how BottleneckMLP effectively (and implicitly) enforces IB by effecting latent space representation dynamics (Section 4.3.2). We empirically validate the effectiveness of BottleneckMLP and current approaches failing to reduce $I(X; Z)$ (Section 4.3.3). We provide further empirical evidence on theory of latent space dynamics in Section 5.

### 4.3.1 WHY EXPLICIT IB FAILS ON GRAPHS

Explicit IB relies on the *Asymptotic Equipartition Property (AEP)*, which holds when data is generated from a stationary, ergodic process with primarily local dependencies. In images or speech, each variable (e.g., a pixel or phoneme) depends mostly on its local neighborhood and is conditionally independent of distant variables given that neighborhood (low global dependence). The AEP theorem (Breiman, 1957) states that for a sequence $X_1, X_2, \ldots$ from distribution $p(x_1, x_2, \ldots)$

$$\lim_{n \to \infty} -\tfrac{1}{n} \log p(x_1, \ldots, x_n) = H(X). \tag{7}$$

Thus, for large $n$, almost all patterns are *typical*. Under local dependence, the joint distribution can be approximated by products of localized conditionals:

$$p(x_1, \ldots, x_n \mid P) \approx 2^{-nH(X|P)} \quad \text{for typical partitions } P. \tag{8}$$

In systems with low global connectivity and primarily local dependencies, conditional probabilities can be factorized locally and averaged via the Central Limit Theorem. Under the *typicality* assumption, a similar effect holds in information theory, causing $I(X; Z)$ and $I(Z; Y)$ to concentrate and enabling reliable estimation from partitions of $p(X, Y)$.

$$I(X; Z) = \mathbb{E}_{X,Z} \left[ \log \frac{p(x \mid z)}{p(x)} \right] = \mathbb{E}_{X,Z} \left[ \sum_i \log \frac{p(x_i \mid \mathcal{N}(x_i), z)}{p(x_i \mid \mathcal{N}(x_i))} \right], \tag{9}$$

$$I(Z; Y) = \mathbb{E}_{Z,Y} \left[ \log \frac{p(y \mid z)}{p(y)} \right] = \mathbb{E}_{Z,Y} \left[ \sum_x p(y \mid x) \prod_i p(x_i \mid \mathcal{N}(x_i), z) - \log p(y) \right], \tag{10}$$

were $\mathcal{N}(x_i)$ denotes the neighborhood of $x_i$. These assumptions justify the use of variational bounds in estimating $I(X; Z)$ and $I(Z; Y)$. However, they break down in graphs, where features are not i.i.d. and nodes are structurally entangled, and where the patterns are not large enough to be *typical*.

**Graphs Break AEP Assumptions** Graphs violate the requisite conditions for AEP. Structural entanglement creates strong global dependencies, the distribution $P(G)$ is not factorized over nodes

or edges (also addressed in (Wu et al., 2020)), and most graphs are too small to exhibit typicality. As a result, variational estimates of $I(X; Z)$ collapse to KL terms that assume i.i.d. sampling, systematically underestimating dependencies. Empirical evidence of these dependencies is provided in Appendix E.3. Thus, explicit IB losses cannot reliably control information flow in GNNs.

### 4.3.2 IMPLICIT IB THROUGH GAUSSIANIZATION

**Implicit Realization of the IB Lagrangian** BottleneckMLP implicitly optimizes the IB objective without requiring variational bounds. Concretely, we obtain:

$$\min_{p(z|x)} -I(Z; Y) + \beta I(X; Z) \rightsquigarrow \begin{cases} I(Z; Y) & \text{is preserved via low-noise important nodes,} \\ I(X; Z) & \text{is reduced via Gaussianization of unimportant nodes.} \end{cases}$$

Decomposing the input representation, $I(X; Z) = I(X; Z_{imp}) + I(X; Z_{unimp})$, and noting that unimportant nodes are conditionally independent of the target given the important nodes, $I(Z_{unimp}; Y \mid Z_{imp}) = 0$, reducing $I(X; Z_{unimp})$ via noise does not hurt prediction. Thus BottleneckMLP achieves the minimal sufficient representation postulated by the IB principle, while avoiding the weaknesses of explicit IB loss functions in graphs.

**Selective Forgetting of Information** Our approach achieves compression by selectively forgetting information about unimportant nodes ($I(X; Z)$ minimization) while preserving structure around important ones. Gaussianization serves as a natural mechanism for information loss. By the Central Limit Theorem, repeated independent noise-injection drives convergence towards Gaussianity (since the embeddings aggregate many independent perturbations). For each unimportant node $i$ we define:

$$Z_i = f_i(X) + t_i \epsilon_i, \quad \epsilon_i \sim \mathcal{N}(0, I_d), \quad t_i := \frac{\sigma}{\alpha_i}, \ \alpha_i \in (0, 1), \tag{11}$$

where $f_i(X) \in \mathbb{R}^d$ is the output of the encoder, and $\sigma$ is a fixed hyperparameter, and and the noise terms $\{\epsilon_i\}$ are sampled independently at each forward pass. We assume $Z_i \mid X$ admits a smooth density with finite Fisher information.

Below we formalize the theoretical groundings of BottleneckMLP. We refer the reader to B for the proofs of Lemma 1 and Lemma 2.

**Lemma 1 (Monotonicity of Conditional Entropy)** *Let $Z_i = f_i(X) + \sqrt{t_i} N_i$ as above. Then the conditional entropy of $Z_i$ given $X$ satisfies the multivariate De Bruijn identity:*

$$\frac{d}{dt_i} H(Z_i \mid X) = \frac{1}{2} \text{Tr}(J(Z_i \mid X)) \geq 0,$$

*where $J(Z_i \mid X)$ is the Fisher information matrix. Therefore conditional entropy $H(Z_i \mid X)$ monotonically increases.*

**Lemma 2 (Bounded Marginal Entropy)** *The marginal entropy of $Z_i$ is bounded by:*

$$H(Z_i) \leq \frac{1}{2} \log\Big( (2\pi e)^d \det(\text{Cov}(f_i(X)) + t_i I_d) \Big).$$

**Theorem 1 (Mutual Information Reduction)** *As noise variances $\{t_i\}_{i \in U}$ increase: (1) $I(X; Z_{\text{unimp}})$ decreases monotonically, and (2) $\lim_{\min_{i \in \text{unimp}} t_i \to \infty} I(X; Z_{\text{unimp}}) = 0$.*

**Proof**

(1) By the chain rule for MI, $I(X; Z) = I(X; Z_{\text{unimp}}) + I(X; Z_{\text{imp}} \mid Z_{\text{unimp}})$. For the unimportant term, $I(X; Z_{\text{unimp}}) = H(Z_{\text{unimp}}) - H(Z_{\text{unimp}} \mid X)$. By Lemma 1, $H(Z_{\text{unimp}} \mid X)$ increases monotonically with $\{t_i\}$; by Lemma 2 the marginal entropy $H(Z_{\text{unimp}})$ is bounded above. Consequently $I(X; Z_{\text{unimp}})$ decreases as the noise variances increase.

(2) As $t_i \to \infty$ for $i \in \text{unimp}$, the noise dominates the signal: $Z_i \approx t_i \epsilon_i$. Since $\epsilon_i$ is independent of $X$, we get $I(X; Z_i) \to 0$ for each $i \in \text{unimp}$. For the joint MI, note that while embeddings $\{f_i(X)\}$ may be correlated, the noise terms $\{\epsilon_i\}$ are independent. As $t_i \to \infty$, we have $Z_{\text{unimp}} \approx [t_1 \epsilon_1, \ldots, t_{|\text{unimp}|} \epsilon_{|\text{unimp}|}]$ where the $\epsilon_i$ are independent of $X$, giving $I(X; Z_{\text{unimp}}) \to 0$.

**Stochastic Relaxation** For DNNs, after initial fitting, gradient noise induces stochastic relaxation where the network implicitly maximizes conditional entropy $H(Z|X)$, minimizing $I(X;Z) = H(Z) - H(Z|X)$ without explicit regularization (Tishby & Zaslavsky, 2015; Tishby et al., 2000). In the graph setting, BottleneckMLP is the key architectural component that elicits this phenomenon.

### 4.3.3 BOTTLENECKMLP EFFECTIVELY ENFORCES IB DYNAMICS

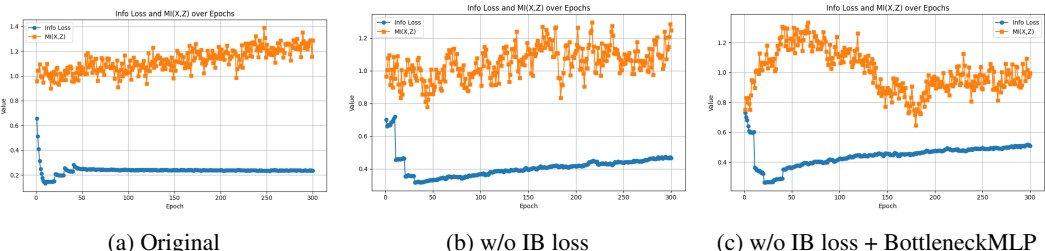

|                    |                      |                                          |
|:------------------:|:--------------------:|:----------------------------------------:|
| (a) Original       | (b) w/o IB loss      | (c) w/o IB loss + BottleneckMLP          |

Figure 2: Mutual information $I(X;Z)$ (orange) and IB Loss (blue) (Equation 5) visualizations over epochs (GSAT on MUTAG). BottleneckMLP enforces compression effectively, whereas explicit IB Loss terms do not. (a) and (b) do not exhibit $I(X;Z)$ minimization. (c) exhibits both the fitting ($I(X;Z) \uparrow$) and compression ($I(Z;Y) \downarrow$) phases consistent with (Tishby & Zaslavsky, 2015). Note that minimization of IB Loss (blue) does not correlate with $I(X;Z)$ (orange). The x-axis shows epochs 0–300. The y-axis is the loss/ MI value in range (0, 1.4).

In Figure 2c, $I(X;Z)$ rises early as task-relevant features are captured, then declines in later epochs, reflecting effective compression. This shows that our architecture encourages forgetting irrelevant details and produces IB dynamics absent in prior methods. HSIC analysis further confirms reduced dependence between $X$ and $Z$ across layers (Appendix E.4). By contrast, explicit IB losses fail: in 2a (GSAT) and 2b (GSAT w/o IB loss), $I(X;Z)$ grows monotonically and IB Loss curves misalign with actual dynamics, highlighting the limits of explicit IB and the advantage of our implicit, architecture-driven approach.

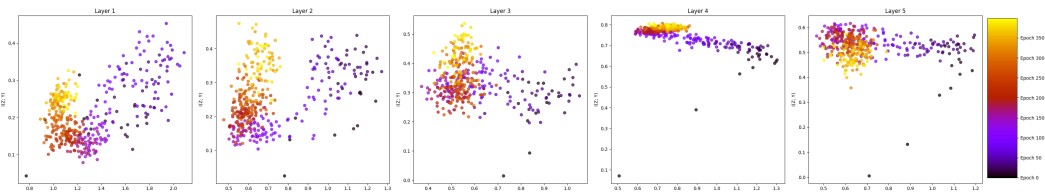

Figure 3: BottleneckMLP enforces IB on graph embeddings in the information plane $I(X;Z_i)$ vs. $I(Z_i;Y)$ over layers, $i$ in range 1–5, and epochs (purple to yellow). Information curve (Tishby & Zaslavsky, 2015) naturally appears with CE minimization. Compression and concentration are more apparent in the later layers, consistent with the theory in Section 4.3.1. X-axis in range (0, 2) for i=1, and (0, 1.3) for i=2–5. Y-axis in range (0,0.5) for i=1–3, (0, 0.8) for i=4, and (0, 0.6) for i=5.

BottleneckMLP effectively reproduces the information plane dynamics originally observed in DNNs by (Tishby & Zaslavsky, 2015), now extended to the graph domain with GNNs, as shown in Figure 3. Both the characteristic fitting and compression phases are clearly visible. This two-phase IB trajectory does not appear in the baseline GNN training with or without Info Loss; we include these plots, respectively, for comparison to Figure 3 in Appendix P. In Figure 3, early layers behave similarly to those in the baseline model, but beginning around Layer 3, clear compression in $(I(X;Z_i))$ emerges as training progresses. Points transition from right (high $(I(X;Z_i))$) to left (compressed representations) while $(I(Z_i;Y))$ becomes more concentrated. By Layers 4 and 5, the classical IB trajectory is clearly visible. This trend aligns with theoretical expectations and with our discussion in Section 4.3.1, demonstrating that BottleneckMLP successfully induces the expected IB behavior in graph representations.

## 5 EXPERIMENTS

Section 5.1 analyzes the distributional properties of node embeddings over training via LNSA and embedding drift to validate our theory on Selective Forgetting of Information given in Section 4.3.2, Section 5.2 reports the improved performance of BottleneckMLP versus baselines and ablations. We focus on ante-hoc IB-based explainers. BottleneckMLP does not impact model efficiency; Appendix N provides this quantitative runtime analysis. For generalizability of BottleneckMLP, we report improved performance across graph tasks and explainer types in Appendix H.

### 5.1 REPRESENTATION DYNAMICS OF BOTTLENECKMLP REFLECT IB

We provide strong mechanistic and empirical evidence that BottleneckMLP induces representational dynamics aligned with the goals of the IB framework, selectively reducing $I(X; Z)$ while preserving $I(Z; Y)$. All experiments corroborate that important node embeddings maintain lower entropy and more structured distributions, while unimportant node embeddings progressively approach Gaussian-like high-entropy distributions. This results in the desired $I(X; Z)$ minimization (4.3.2). We present LNSA and node drift results, and refer the reader to Appendix for convex hull volume (E.2), node linkage (E.3), HSIC (E.4), and additional node drift experiments (F).

**Results on Localized Normalized Space Alignment (LNSA)** Following (Ebadulla et al., 2025), we track how representation geometry evolves across epochs using LNSA, which measures neighborhood alignment across epochs (higher values indicate greater instability) (details in Appendix K). Nodes are grouped by importance: Category 1 (important with important neighbors), Category 2 (important with mixed neighbors), and Category 3 (unimportant with unimportant neighbors). In Figure 4, we observe that Categories 1 and 2 maintain low LNSA over epochs, while Category 3 exhibits high values, reflecting drift toward high-entropy distributions under BottleneckMLP.

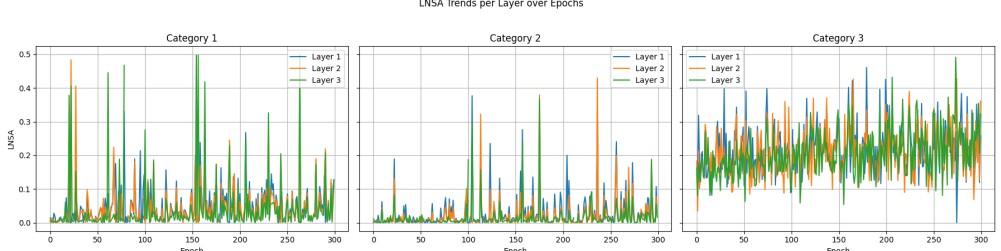

Figure 4: LNSA values for PGIB + BottleneckMLP. Cat 1: important nodes. Cat 2: important nodes with unimportant neighbors. Cat 3: unimportant nodes. Cat 1 and Cat 2 embeddings remain similar, while Cat 3 and has a higher mean LNSA value as embedding structure changes over epochs.

**Results on Embedding Drift** Figure 5 shows embedding drift under different GSAT configurations. Baseline GSAT yields little separation between node types, and removing the IB Loss term causes uniformly unstable drift. With BottleneckMLP, we achieve the intended IB effect in alignment with out theory: important nodes stabilize with low drift, while unimportant nodes continue drifting toward noisier, Gaussian-like distributions. This pattern holds across all datasets (see Appendix E.1). Notably, these effects occur without an explicit IB loss, demonstrating that BottleneckMLP introduces a powerful implicit bottleneck via architectural constraints alone.

### 5.2 BOTTLENECKMLP IMPROVES CLASSIFICATION AND EXPLANATION

We confirm that IB Loss adds value beyond supervised training by comparing baseline ante-hoc, IB-based explainers with/ without the IB Loss, and BottleneckMLP (experimental setup in Appendix C). We evaluate performance on benchmarks using accuracy, explanation AUC-ROC, and Fidelity$\pm$ (Appendix J). We then demonstrate the generalizability of BottleneckMLP across explainer types (post-hoc or ante-hoc and IB-based or non-IB-based) and for graph tasks (subgraph recognition, node classification, graph classification).

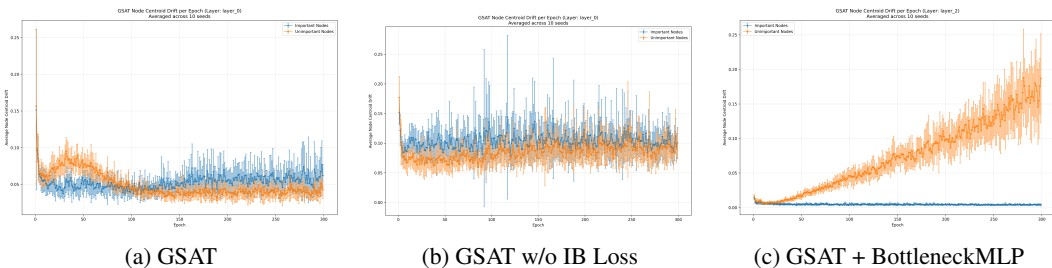

| (a) GSAT | (b) GSAT w/o IB Loss | (c) GSAT + BottleneckMLP |

Figure 5: Drift of important (blue) vs. unimportant (orange) nodes across epochs for: (a) GSAT, (b) GSAT w/o IB Loss, and (c) GSAT with BottleneckMLP. BottleneckMLP alone correctly affects representation dynamics. We see the same plots across models and datasets (in Appendix E).

### 5.3 BottleneckMLP Performance on Ante-hoc IB-Based Explainers

Table 1 shows the ineffectiveness of IB Loss in GSAT and PGIB, where adding BottleneckMLP consistently improves performance. On BA-2Motifs, where IB Loss removal hurts performance, BottleneckMLP recovers and increases accuracy, replicating and surpassing the role of IB Loss.

Table 1: PGIB/ GSAT Classifier Accuracy. BottleneckMLP increases performance over the original explainer method, and when IB Loss is removed.

| | MUTAG | BA-2Motifs | NCI1 | PROTEINS | Benzene | Alkane Carbonyl | Fluorine Carbonyl |
|---|---|---|---|---|---|---|---|
| GSAT | 0.909±0.033 | 0.994±0.006 | 0.689±0.017 | 0.681±0.036 | 1.000 | 0.907±0.151 | 0.809±0.005 |
| GSAT w/o IB Loss | 0.935±0.043 | 0.770±0.224 | 0.743±0.024 | 0.706±0.040 | 1.000 | 0.939±0.026 | 0.958±0.077 |
| GSAT w/o IB Loss + BottleneckMLP | **0.949**±0.010 | **1.000** | **0.802**±0.019 | **0.745**±0.048 | 1.000 | **0.995**±0.006 | **0.987**±0.013 |
| PGIB | 0.904 ± 0.010 | 0.628 ± 0.171 | 0.729 ± 0.024 | 0.729 ± 0.024 | 0.886 ± 0.008 | 0.994 ± 0.002 | 0.989 ± 0.011 |
| PGIB w/o IB Loss | 0.916 ± 0.011 | 0.896 ± 0.145 | **0.774** ± 0.014 | 0.763 ± 0.022 | 0.899 ± 0.006 | 0.0987 ± 0.013 | 0.939 ± 0.002 |
| PGIB w/o IB Loss + BottleneckMLP | **0.925** ± 0.009 | **0.963** ± 0.016 | 0.753 ± 0.019 | **0.792** ± 0.018 | **0.900** ± 0.004 | **0.995** ± 0.000 | **0.993** ± 0.003 |

Table 2: TGIB Link Prediction AP and Explanation AUC/ROC. BottleneckMLP improves performance across datasets, where removal of IB Loss also increases performance

| | **Link Prediction (AP)** | | | **Explanation AUC/ROC** | | |
|---|---|---|---|---|---|---|
| **Model** | CanParl | USLegis | Wikipedia | CanParl | USLegis | Wikipedia |
| TGIB | 0.789 | 0.828 | 0.991 | 0.588 | 0.673 | 0.983 |
| TGIB w/o IB Loss | 0.814 | 0.763 | 0.994 | 0.629 | 0.541 | 0.989 |
| TGIB w/o IB Loss + Bottleneck MLP | **0.82** | **0.843** | **0.994** | **0.632** | **0.703** | **0.989** |

In Table 2, BottleneckMLP improves Average Precision (main metric) and AUC-ROC for TGIB. Performance gain is largest on USLegis, while Wikipedia and CanParl have comparable or slightly better performance. Removing IB Loss boosts performance on Wikipedia and CanParl but reduces it on USLegis; adding BottleneckMLP recovers this loss and exceeds the baseline. Classifier accuracy is not reported on original TGIB paper, we include it in the Appendix H.

Table 3: BottleneckMLP consistently improves Fidelity $Fid^+ \uparrow$ / $Fid^- \downarrow$ on PGIB.

| Dataset | PGIB | PGIB w/o IB Loss | PGIB + BottleneckMLP |
|---|---|---|---|
| MUTAG | 0.750 ± 0.079 / 0.588 ± 0.204 | 0.719 ± 0.083 / 0.516 ± 0.192 | **0.762** ± 0.071 / **0.383** ± 0.254 |
| BA-2Motifs | 0.825 ± 0.159 / 0.492 ± 0.149 | 0.829 ± 0.160 / 0.479 ± 0.143 | **0.975** ± 0.079 / **0.400** ± 0.242 |
| NCI1 | 0.451 ± 0.124 / 0.523 ± 0.156 | 0.524 ± 0.136 / 0.546 ± 0.125 | **0.771** ± 0.162 / **0.478** ± 0.214 |
| PROTEINS | 0.639 ± 0.024 / 0.602 ± 0.125 | 0.654 ± 0.028 / 0.604 ± 0.035 | **0.658** ± 0.018 / **0.592** ± 0.087 |
| Benzene | 0.508 ± 0.016 / 0.042 ± 0.000 | 0.516 ± 0.022 / 0.043 ± 0.006 | **0.636** ± 0.049 / **0.042** ± 0.000 |
| Alkane Carbonyl | 0.579 ± 0.038 / 0.180 ± 0.076 | 0.915 ± 0.035 / 0.101 ± 0.053 | **0.967** ± 0.017 / **0.093** ± 0.032 |
| Fluorine Carbonyl | 0.767 ± 0.027 / 0.149 ± 0.050 | 0.794 ± 0.069 / 0.138 ± 0.045 | **0.916** ± 0.033 / **0.080** ± 0.029 |

Table 3 reports fidelity metrics for PGIB on MUTAG and BA-2Motifs (the only datasets with ground-truth explanations). Adding BottleneckMLP consistently improves $Fid^+$ and reduces $Fid^-$, yielding higher-quality explanations. Table 2 reports improved AUC-ROC scores across datasets for

TGIB with BottleneckMLP. AUC-ROC scores and information planes for GSAT variants are provided in Appendix F. Visual comparisons of explanatory subgraphs appear in Appendix D.

## 5.4 GENERALIZATION OF BOTTLENECKMLP ACROSS EXPLAINABILITY METHODS

We included all existing ante-hoc graph explainers that use variational IB objectives as our baselines. For generalizability, we tested BottleneckMLP on subgraph recognition (Yu et al., 2022), post-hoc node classification (Luo et al., 2020b), non-IB-based ante-hoc explainers (Wu et al., 2022; Lu et al., 2024), and non-IB-based post-hoc explainers (Wang et al., 2023; 2021). Across explanation models and tasks, BottleneckMLP consistently exhibits superior performance.

BottleneckMLP improves subgraph recognition accuracy in VGIB (Yu et al., 2022) by $3.5 - 62.5\%$ depending on dataset (Table 15). For post-hoc node classification, BottleneckMLP improves PG-Explainer (Luo et al., 2020b) accuracy by $1.1 - 4.9\%$, and explanation AUC-ROC by $0.6 - 0.8\%$ for BA-Shapes and TreeGrids, while AUC-ROC is $0.3\%$ lower for TreeCycles. However, for PG-Explainer, removing Entropy Loss is extremely detrimental to performance, and BottleneckMLP is able to fully recover and slightly improve this performance (Table 17). BottleneckMLPsignificantly improves AUC, $Fid^+$, and $Fid^-$ for ProxyExplainer (Chen et al., 2024) across all four datasets except BA-2Motifs on AUC and $Fid^+$, and Fluoride Carbonyl on $Fid^-$. ReFine (Wang et al., 2021) with BottleneckMLP has comparable performance to the baseline ReFine model on BA3 and MNIST datasets, fully recovering the function of the fidelity loss in all cases. It improves ACC-AUC on MUTAG by $19.9\%$. On V-InFoR (Wang et al., 2023) (post-hoc, GIB-based graph classification explainer), and on GOAt (Lu et al., 2024) (ante-hoc non-IB-based graph classification explainer), BottleneckMLP improves all metrics across all datasets (Tables 18, 21). On DIR (Wu et al., 2022), another ante-hoc non-IB-based graph classification explainer, BottleneckMLP improves training accuracy by up to $24.2\%$, validation accuracy by up to $8.2\%$, and lowers confounding accuracy by $5.1\%$. It has comparable, yet slightly lower performance on the other metrics; however, BottleneckMLP is able to recover the performance increase caused when we remove the Info Loss terms, showing that it implicitly causes the same effects as does the explicit loss in DIR. Further explanation discussion, and results for these individual models is in Appendix H.

## 5.5 SENSITIVITY ANALYSIS OF $\sigma$ AND CONVERGENCE RESULTS

The hyperparameter $\sigma$ in Equation 6 controls the strength of the node-embedding perturbation. A full sensitivity analysis is provided in Appendix S. $\sigma$ must be tuned per model–dataset pair to obtain the optimal value due to varying training dynamics. We further observe that BottleneckMLP accelerates convergence by inducing the desired representation dynamics throughout training. Justification and empirical results for this speed-up are reported in Table 26 in Appendix O.

## 6 DISCUSSION AND FUTURE DIRECTIONS

BottleneckMLP is an architectural module that implicitly enforces IB principle without loss terms. Our model-agnostic method relies on two components: Importance-Weighted Gaussianization and Progressive Compression via MLP layers. We both theoretically and empirically demonstrate the effectiveness of these components, and show that explicit IB losses are less effective.

Formalizing the conditions under which implicit bottlenecks arise remains an open theoretical challenge, which is a limitation of our work. A deeper information-theoretic analysis of layer-wise compression under graph message passing could help unify our observations with general learning theory. Another limitation and possible future direction is identifying optimal BottleneckMLP architectures for specific datasets and explainers. We offer a representational geometric lens via LNSA; future work can study how representation dynamics impact generalization and robustness in graph learning. Promoting Gaussianity Eftekhari & Papyan (2025) or disentanglement Pan et al. (2020) in hidden representations may amplify IB effects. Building on the inductive biases observed in Graph Neural Tangent Kernels (GNTKs) Du et al. (2019), future work can explore how inducing normality may contribute to stable, explainable, and noise-resilient representations in GNNs.

## 7 REPRODUCIBILITY STATEMENT

All datasets used are available for download from the original source, linked in Appendix C. All code needed to reproduce experiments and generate all figures within the paper is available in the supplementary material, and experimental setup information is also in Appendix C. Follow the guidelines on the supplementary material README pages to reproduce the performance results, as well as the plots presented throughout the paper.

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

APPENDIX: TECHNICAL DETAILS AND SUPPLEMENTARY MATERIAL

TABLE OF CONTENTS

## A  LIST OF SYMBOLS

## B  PROOFS OF LEMMA 1 AND LEMMA 2

**Proof of Lemma 1**  *$J(\cdot)$ is positive semi-definite, therefore trace is nonnegative. Thus $H(Z_i \mid X)$ is monotonically increasing in the noise variance $t_i$. The same argument applies jointly to the vector $Z_{\mathrm{unimp}}$, since independent Gaussian noise is injected into each unimportant node.*

**Proof of Lemma 2**  *By the law of total covariance,*

$$\mathrm{Cov}(Z_i) = \mathbb{E}[\mathrm{Cov}(Z_i \mid X)] + \mathrm{Cov}(\mathbb{E}[Z_i \mid X]) = t_i I_d + \mathrm{Cov}(f_i(X)).$$

*The Gaussian distribution maximizes entropy among all distributions with fixed covariance, $\Sigma_{Z_i}$, which yields the bound.*

## C  EXPERIMENTAL SETUP

All experiments were implemented using `PyTorch Geometric` and run on either CPU or NVIDIA H200 GPUs. Unless otherwise stated, the following hyperparameters were used:

Tables 6 and Table 7 give an overview of the dataset statistics used within the paper.

| Symbol | Description |
|---|---|
| $G$ | Input graph |
| $G_S$ | Explanation subgraph |
| $X$ | Input feature vector (node embeddings of $G$ after graph layers) |
| $Z$ | Learned hidden representation |
| $Z_{imp}$ | Learned hidden representations for important nodes |
| $Z_{unimp}$ | Learned hidden representations for unimportant nodes |
| $Z_i$ | Noisy hidden representation for node $i$ |
| $Y$ | Ground truth label |
| $H(X)$ | Entropy of r.v. X |
| $I(X;Z)$ | Mutual information between X and Z |
| $\beta$ | Lagrangian multiplier for the IB functional |
| $g_\phi(G)$ | Subgraph extractor |
| $P_\phi(G_S|G)$ | Distribution of subgraph $G_S$ outputted by $g_\phi(G)$ |
| $\mathcal{N}(x_i)$ | Neighborhood of $x_i$ |
| $f_i$ | Encoder yielding node/ edge embeddings |
| $t_i$ | Noise magnitude, inversely proportional to importance |
| $\epsilon_i$ | Gaussian noise vector |
| $I_d$ | d × d identity matrix |
| $\alpha_i$ | Importance score of node $i$ |
| $\sigma$ | Fixed noise scale; hyperparameter |
| $J(\cdot)$ | Fischer information matrix |
| $Tr(\cdot)$ | Trace of a matrix |
| det | Matrix determinant |
| Cov | Covariance |
| $d$ | Dimension of embedding space |
| $G_p$ | Prototype graph in PGIB |
| $Q(G_S)$ | Assumed prior for $G_S$ for the KL variational upper bound |
| $p(x_1, x_2, ..., x_n)$ | Joint distribution of random variables |
| $\mathcal{N}(x_i)$ | The local neighborhood on which r.v. $x_i$ depends |
| $\Delta I_X^k$ | Partial compression at layer $k$ |
| $h_v^k$ | Embedding of node $v$ in $k^{th}$ layer |
| 'IB Loss' | Loss term explicitly which explicitly enforces IB principle |

Table 4: Source code links of baseline models

| Method | Source code |
|---|---|
| GSATMiao et al. (2022) | https://github.com/Graph-COM/GSAT/tree/main |
| PGIBSeo et al. (2024b) | https://github.com/sang-woo-seo/PGIB |
| TGIBSeo et al. (2024a) | https://github.com/sang-woo-seo/TGIB |

Table 5: Summary of model hyperparameters.

| Model | Learning Rate | Weight Decay | Batch Size | # Epochs | # Random Seeds |
|---|---|---|---|---|---|
| GSATMiao et al. (2022) | $10^{-3}$ | 0.0 | 128 | 100 | 10 |
| PGIBSeo et al. (2024b) | $5^{-3}$ | 0.0 | 128 | 300 | 10 |
| TGIBSeo et al. (2024a) | $10^{-5}$ | 0.0 | 200 | 10 | 10 |

Table 6: Overview of graph classification datasets used in experiments on GSAT and PGIB.

| Dataset | #Graphs | #Classes | Avg. # Nodes | Avg. # Edges | Node Labels | Edge Labels | Node Attr. (Dim.) | Edge Attr. (Dim.) |
|---|---|---|---|---|---|---|---|---|
| MUTAG Debnath et al. (1991) | 188 | 2 | 17.93 | 19.79 | + | + | - | - |
| NCI1 Morris et al. (2020) | 4110 | 2 | 29.87 | 32.30 | + | - | - | - |
| PROTEINS Morris et al. (2020) | 1113 | 2 | 39.06 | 72.82 | + | - | + | - |
| BA_2Motifs Luo et al. (2020b) | 1000 | 2 | 25 | 51.39 | - | - | - | - |

Table 7: Overview of node classification datasets used in experiments on TGIB.

| Dataset | Domain | #Nodes | #Edges | #Edge Features | Duration |
|---|---|---|---|---|---|
| WikipediaKumar et al. (2019) | Social | 9,227 | 157,474 | 172 | 1 month |
| CanParlHuang et al. (2020) | Politics | 734 | 74,478 | 1 | 14 years |
| USLegisHuang et al. (2020) | Politics | 225 | 60,396 | 1 | 12 terms |

# D EXPLANATORY SUBGRAPH VISUALIZATION

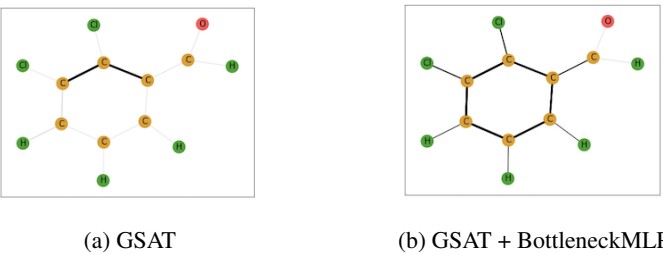

(a) GSAT      (b) GSAT + BottleneckMLP

Figure 6: Visualization of explanatory subgraphs for MUTAG dataset. (a) GSAT baseline, and (b) GSAT enhanced with BottleneckMLP. BottleneckMLP correctly identifies the carbon ring for non-mutagenic molecules.

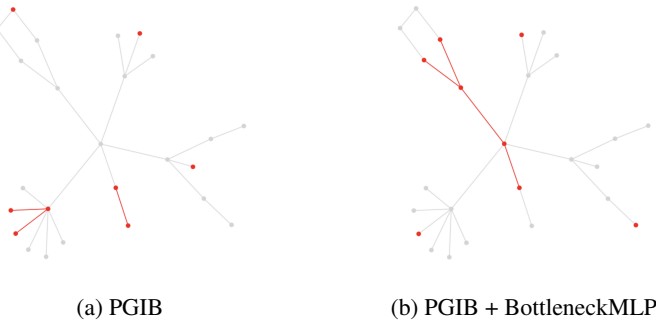

(a) PGIB      (b) PGIB + BottleneckMLP

Figure 7: Visualization of explanatory subgraphs for BA-2Motifs dataset. (a) PGIB baseline, and (b) PGIB enhanced with BottleneckMLP. BottleneckMLP correctly identifies the cycle motif.

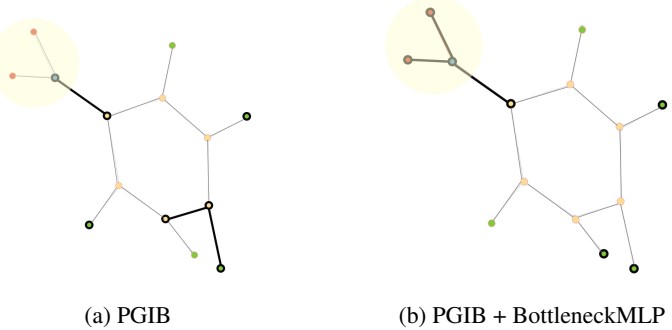

(a) PGIB      (b) PGIB + BottleneckMLP

Figure 8: Visualization of explanatory subgraphs for MUTAG dataset. (a) PGIB baseline, and (b) PGIB enhanced with BottleneckMLP. BottleneckMLP successfully identifies the NO2 group (yellow shaded region), which is the ground truth explanation for mutagenic molecules in MUTAG.

# E    REPRESENTATION DYNAMICS ACROSS LAYERS

Supplementary to the NSA and node drift results in Section 5.1, in this section we include these results for additional models, and we study the representation dynamics of the important versus unimportant node embeddings across training epochs from several lenses including (1) node drift, (2) convex hull volume, (3) average linkage distance. Each of these further corroborates the effectiveness of BottleneckMLP in implicitly enforcing IB as an architectural primitive which effects the representation dynamics over training.

## E.1    NODE DRIFT

Table 8: Difference between unimportant and important node drift (Unimp − Imp) for GSAT. With BottleneckMLP, we validate across datasets that important nodes stabilize and unimportant nodes drift to higher-entropy representations.

| Architecture / Configuration | MUTAG | BA-2Motifs | NCI1 | PROTEINS |
|---|---|---|---|---|
| GSAT | $-0.02 \pm 0.007$ | $0.006 \pm 0.006$ | $-0.004 \pm 0.006$ | $0.002 \pm 0.001$ |
| GSAT w/o IB Loss | $-0.018 \pm 0.007$ | $-0.003 \pm 0.006$ | $0.028 \pm 0.011$ | $\mathbf{0.004 \pm 0.013}$ |
| GSAT w/o IB Loss + BottleneckMLP | $\mathbf{0.07 \pm 0.009}$ | $\mathbf{0.02 \pm 0.006}$ | $\mathbf{0.04 \pm 0.054}$ | $-0.003 \pm 0.015$ |

Table 9: Difference between unimportant and important node drift (Unimp − Imp) for PGIB. With BottleneckMLP, we validate across datasets that important nodes stabilize and unimportant nodes drift to higher-entropy representations.

| Architecture / Configuration | MUTAG | BA-2Motifs | NCI1 | PROTEINS |
|---|---|---|---|---|
| PGIB | $0.087 \pm 1.014$ | $0.126 \pm 0.240$ | $0.227 \pm 0.127$ | $0.129 \pm 0.0531$ |
| PGIB w/o IB Loss | $-0.205 \pm 0.169$ | $0.186 \pm 0.457$ | $-0.116 \pm 0.128$ | $0.294 \pm 0.095$ |
| PGIB w/o IB Loss + BottleneckMLP | $\mathbf{0.546 \pm 0.111}$ | $\mathbf{0.214 \pm 0.049}$ | $\mathbf{0.415 \pm 0.272}$ | $\mathbf{0.937 \pm 0.563}$ |

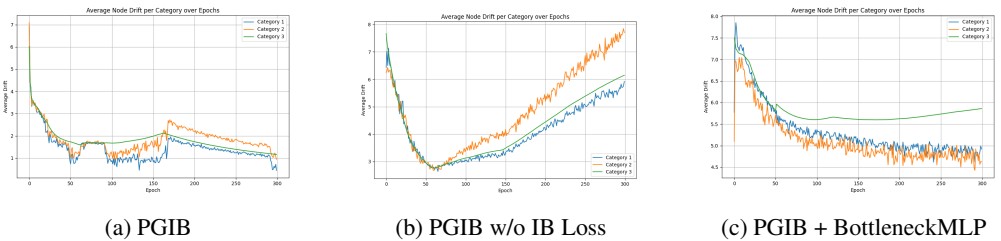

| (a) PGIB | (b) PGIB w/o IB Loss | (c) PGIB + BottleneckMLP |
|---|---|---|

Figure 9: Drift of important vs. unimportant nodes across models: (a) PGIB baseline, (b) PGIB without Information Loss, and (c) PGIB with BottleneckMLP. The blue line represents the average drift over epochs of the category 1 nodes, the orange line is that of category 2 nodes, and the green is that of category 3 nodes. We see that BottleneckMLP effectively enforces drift/ forgetting of unimportant nodes, as evident by their increased drift in latent space across epochs.

## E.2    CONVEX HULL

To further support our hypothesis that the BottleneckMLP influences representation structure locally—particularly around important nodes—we analyze the convex hull volumes and average linkage distances of node embeddings over training.

As described earlier, we project node embeddings into 3D using PCA and compute the convex hull volume for each category across training epochs. Our findings, visualized in Figure 10, indicate a clear trend: Category 3 nodes (unimportant) occupy significantly larger and more variable convex hulls compared to Category 1 nodes (important). This increasing spatial dispersion suggests that unimportant nodes drift more in embedding space as training progresses, while important nodes remain compact and tightly clustered—consistent with more stable local structure.

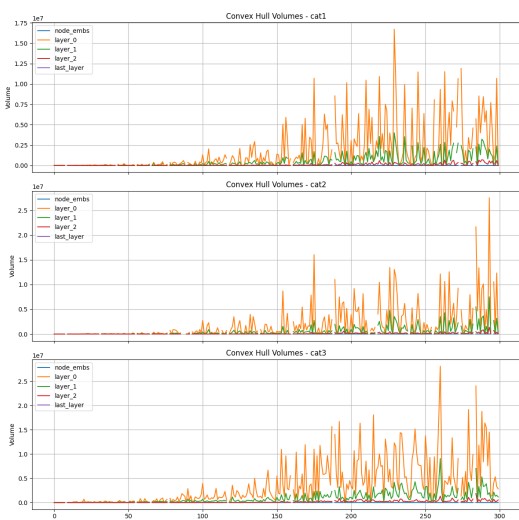

Figure 10: Convex hull volumes increase across categories. The x-axis spans epochs 0 to 300. The y-axis range is (0.0, 1.75e7) for Category 1 (top plot), and is (0.0, 3.0e7) for Categories 2 and 3 (middle and bottom plot). For all layers (most significantly for earlier layers), the convex hull volume increases across categories with Category 1 having the smallest convex hull volume, and Category 3 with the largest convex hull volume.

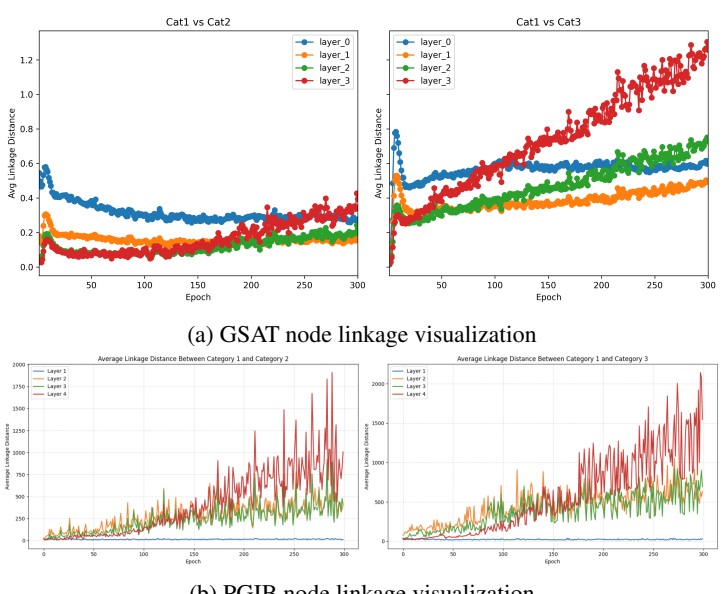

(a) GSAT node linkage visualization

(b) PGIB node linkage visualization

Figure 11: Comparison of distances from Category 1 nodes to Category 3 nodes. Higher drift in the embedding space of unimportant nodes increases these distances. We also present the importance of node dependencies in graphs, Category 3 nodes drift far more than Category 2 nodes, even though they are both unimportant

### E.3 NODE LINKAGE DISTANCE

To complement this analysis, we compute the average linkage distance between node embeddings within each category. This metric quantifies the average pairwise distance between points in a cluster, offering an alternative view of intra- and inter-category embedding dynamics. The results, summarized in Table 11, closely mirror the trends observed in the convex hull and LNSA analyses. Category 1 nodes consistently exhibit low average intra-category linkage distances, confirming strong

internal cohesion. Category 3 nodes show a significant increase in linkage distance, especially at deeper layers and later epochs, reinforcing the notion of representational drift in less important regions. Category 2 nodes, which represent the neighbors of important nodes, exhibit average linkage values that lie between those of Categories 1 and 3, but are closer to Category 1. This suggests that these nodes remain structurally and representationally aligned with important nodes due to their direct connections and dependency.

We also report the average inter-category linkage distances in Table 10. Notably, the largest inter-cluster distances are observed between Category 1 and Category 3, and between Category 2 and Category 3. In contrast, the average distance between Category 1 and Category 2 embeddings remains relatively low, highlighting their continued proximity in latent space. These patterns further confirm the embedding drift of unimportant nodes away from critical substructures.

Table 10: Inter-Cluster Average Linkage for Categories and Layers. Categories 2 and 3 have higher inter-cluster average linkage distances than Category 1 node embeddings. This shows that the unimportant node embeddings become less more dispersed from the important node embeddings.

| Layer | Cat1 - Cat2 | Cat1 - Cat3 | Cat2 - Cat3 |
|---|---|---|---|
| Layer 1 | 14.955 | 22.934 | 23.155 |
| Layer 2 | 285.214 | 450.272 | 451.345 |
| Layer 3 | 236.287 | 383.437 | 383.836 |
| Layer 4 | 409.867 | 568.589 | 568.324 |

Table 11: Intra-Cluster Average Linkage for Categories and Layers. Important node embeddings become more clustered, while unimportant node embeddings become more disperse.

| Layer | Cat1 | Cat2 | Cat3 |
|---|---|---|---|
| Layer 1 | 12.734 | 14.277 | 23.000 |
| Layer 2 | 234.417 | 256.448 | 450.396 |
| Layer 3 | 197.862 | 215.450 | 383.092 |
| Layer 4 | 351.402 | 387.511 | 563.660 |

The results reported in Table 11 for the average linkage distance between a Category 1 node embedding, and the closest node embedding from Category 2, and Category 3, respectively, are visualized in Figure 11.

Together, these metrics validate our claim that important nodes form stable neighborhoods in embedding space, while unimportant nodes undergo more drift. Importantly, this localized drift is not easily observable through global metrics, but becomes clearly evident through localized geometric and clustering analyses. Overall, this set of geometric and clustering-based analyses underscores the model's localized influence on representation learning, and validates our claim: the BottleneckMLP selectively shapes local representations, stabilizing meaningful substructures while allowing greater flexibility and dispersion in the remainder of the graph.

### E.4   HSIC RESULTS

Hilbert-Schmidt Independence Criterion (HSIC) is a kernel-based method for measuring statistical dependence between variables Gretton et al. (2005). We use HSIC to quantify the dependence between the output of GCN layers ($X$) and the final node embeddings before classification ($Z$), as an alternative to mutual information. Lower HSIC values indicate greater independence, helping us assess how much information $Z$ retains from $X$.

We observe that lowest values of dependence between input graph $G$ and learned latent representation of the explanation $G_S$ are achieved with BottleneckMLP. (Table 12)

## F   AUC-ROC

Table 13 shows that explanation AUC-ROC remains consistently high across all GSAT variants, even when the information loss term is removed. The differences are not statistically significant, suggest-

Table 12: HSIC values across datasets for GSAT. Lower values indicate more independence. BottleneckMLP reports lowest HSIC values for 3/4 datasets.

| Architecture / Configuration | BA-2Motifs | MUTAG | PROTEINS | NCI1 |
|---|---|---|---|---|
| GSAT | $3.39 \times 10^{-3}$ | $6.23 \times 10^{-3}$ | $2.95 \times 10^{-3}$ | $6.98 \times 10^{-3}$ |
| GSAT w/o IB Loss | $\mathbf{1.11 \times 10^{-3}}$ | $1.01 \times 10^{-2}$ | $2.89 \times 10^{-3}$ | $6.87 \times 10^{-3}$ |
| GSAT w/o IB Loss + BottleneckMLP | $1.48 \times 10^{-3}$ | $\mathbf{3.89 \times 10^{-3}}$ | $\mathbf{2.53 \times 10^{-3}}$ | $\mathbf{6.37 \times 10^{-3}}$ |

Table 13: GSAT AUC-ROC. BottleneckMLP has comparable explanation quality.

| | MUTAG | BA-2Motifs |
|---|---|---|
| GSAT | $0.995 \pm 0.03$ | $0.988 \pm 0.01$ |
| GSAT w/o IB Loss | $0.998 \pm 0.04$ | $0.996 \pm 0.00$ |
| GSAT w/o IB Loss + Bottleneck MLP | $0.987 \pm 0.01$ | $0.982 \pm 0.01$ |

ing that explicit information loss is not essential for generating high-quality subgraphs. Notably, our BottleneckMLP matches this performance, preserving explanatory power without additional loss terms. Figure 12 further reinforces this conclusion. In the information plane, GSAT with BottleneckMLP (Figure 12b) achieves higher mutual information between the explanatory subgraph and the label, $I(G_s, y)$, without increasing $I(G, G_s)$. This demonstrates that our approach improves both label relevance and compression.

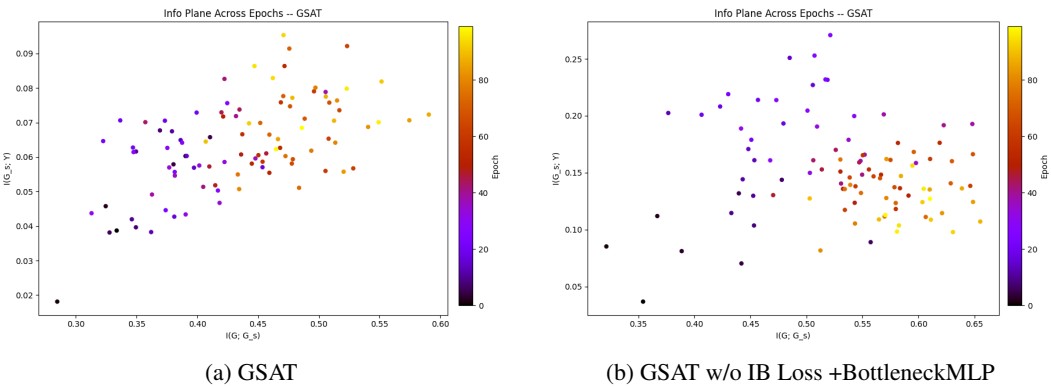

(a) GSAT                    (b) GSAT w/o IB Loss +BottleneckMLP

Figure 12: Comparison of $I(G, G_s)$ vs. $I(G_s, Y)$ across training epochs for the original GSAT model, and our model with a BottleneckMLP with fully connected layers. BottleneckMLP elicits both decreased $I(X; Z)$ and increased $I(Z; Y)$, as compared to original GSAT.

## G  TESTING BOTTLENECKMLP ON A VARIETY OF ARCHITECTURES

We tested multiple fully-connected architectures inserted after the GNN layers of the existing explainer models. All BottleneckMLP architectures use ReLU activations. Determining the optimal architecture of the BottleneckMLP for a given dataset/ explainer model is analogous to hyperparameter tuning.

## H  GENERALIZATION OF BOTTLENECKMLP ACROSS GRAPH TASKS AND EXPLAINER TYPES

We used all existing ante-hoc graph explainers that use variational IB objectives as our baselines. For generalizability, we tested BottleneckMLP on subgraph recognition Yu et al. (2022), post-hoc node classification Luo et al. (2020b), a non-IB-based ante-hoc explainer Wu et al. (2022), and non-IB-based post-hoc explainers Wang et al. (2023); Lu et al. (2024); Wang et al. (2021).

Table 14: PGIB/ GSAT Test Accuracy for various BottleneckMLP Architectures.

|  | MUTAG | BA-2Motifs | NCI1 | PROTEINS |
|---|---|---|---|---|
| GSAT w/o IB Loss + Bottleneck MLP (64-64-64) | $0.943 \pm 0.019$ | $0.959 \pm 0.066$ | $0.799 \pm 0.018$ | $0.750 \pm 0.048$ |
| GSAT w/o IB Loss + Bottleneck MLP (64-48-32) | $0.959 \pm 0.02$ | $0.995 \pm 0.18$ | $0.798 \pm 0.01$ | $0.749 \pm 0.058$ |
| GSAT w/o IB Loss + Bottleneck MLP (64-32) | $0.942 \pm 0.010$ | $0.930 \pm 0.131$ | $0.797 \pm 0.018$ | $0.729 \pm 0.056$ |
| GSAT w/o IB Loss + Bottleneck MLP (64-32-16) | $0.942 \pm 0.013$ | $0.854 \pm 0.162$ | $0.796 \pm 0.019$ | $0.747 \pm 0.043$ |
| GSAT w/o IB Loss + Bottleneck MLP (64-128) | $0.926 \pm 0.037$ | $0.904 \pm 0.152$ | $0.788 \pm 0.025$ | $0.732 \pm 0.067$ |
| PGIB w/o IB Loss + Bottleneck MLP (128-128-128) | $0.928 \pm 0.011$ | $0.967 \pm 0.009$ | $0.771 \pm 0.009$ | $0.769 \pm 0.015$ |
| PGIB w/o IB Loss + Bottleneck MLP (128-96-72) | $0.918 \pm 0.016$ | $0.947 \pm 0.033$ | $0.769 \pm 0.010$ | $0.784 \pm 0.015$ |
| PGIB w/o IB Loss + Bottleneck MLP (128-64) | $0.923 \pm 0.026$ | $0.939 \pm 0.055$ | $0.763 \pm 0.017$ | $0.0766 \pm 0.036$ |
| PGIB w/o IB Loss + Bottleneck MLP (128-64-32) | $0.923 \pm 0.014$ | $0.940 \pm 0.040$ | $0.766 \pm 0.010$ | $0.782 \pm 0.014$ |
| PGIB w/o IB Loss + Bottleneck MLP (128-256) | $0.919 \pm 0.019$ | $0.959 \pm 0.028$ | $0.753 \pm 0.016$ | $0.779 \pm 0.018$ |

## H.1 BOTTLENECKMLP FOR SUBGRAPH RECOGNITION

For VGIB Yu et al. (2022), we removed the MI loss penalty term, and added BottleneckMLP. For VGIB in Table 15, the default model hidden dimension $h = 16$.

Table 15: VGIB Test Prediction Accuracy $\pm$ Std Dev. Variants of VGIB with BottleneckMLP have the highest accuracy across all datasets as compared with baseline VGIB, and VGIB without IB Loss.

| Model Variant | MUTAG | PROTEINS | AIDS | NCI1 |
|---|---|---|---|---|
| VGIB (normal) | $0.423 \pm 0.147$ | $0.573 \pm 0.036$ | $0.418 \pm 0.153$ | $0.496 \pm 0.069$ |
| VGIB (noinfo) | $0.573 \pm 0.167$ | $0.569 \pm 0.031$ | $0.335 \pm 0.207$ | $0.494 \pm 0.043$ |
| VGIB + BottleneckMLP ($h - \frac{h}{4} - h$) | $\mathbf{0.607} \pm 0.232$ | $0.556 \pm 0.075$ | $0.530 \pm 0.345$ | $\underline{0.500} \pm 0.062$ |
| VGIB + BottleneckMLP (128-32) | $0.556 \pm 0.231$ | $\mathbf{0.593} \pm 0.002$ | $\mathbf{0.679} \pm 0.313$ | $0.474 \pm 0.019$ |
| VGIB + BottleneckMLP (64-48-32) | $\underline{0.594} \pm 0.233$ | $\underline{0.592} \pm 0.002$ | $\underline{0.576} \pm 0.238$ | $\mathbf{0.561} \pm 0.062$ |

## H.2 BOTTLENECKMLP FOR POST-HOC NODE CLASSIFICATION

For PGExplainer Luo et al. (2020b), we removed size and entropy losses, kept only cross-entropy, and added BottleneckMLP to the node classifier for implicit compression. Table 16 shows that we achieve better generalization and accuracy in node classifiaction when BottleneckMLP is added.

Table 16: BottleneckMLP improves GNN Node Classification Accuracy

|  | BAShapes | TreeCycles | TreeGrids |
|---|---|---|---|
| GCN | $\underline{0.954} \pm 0.009$ | $0.916 \pm 0.037$ | $\underline{0.806} \pm 0.075$ |
| GCN + BottleneckMLP (48-30) | $\mathbf{0.981} \pm 0.013$ | $\underline{0.959} \pm 0.013$ | $\mathbf{0.815} \pm 0.111$ |
| GCN + BottleneckMLP (30) | $0.979 \pm 0.013$ | $\mathbf{0.961} \pm 0.008$ | $0.782 \pm 0.162$ |

Table 16 shows that without the entropy loss, PGExplainer suffers a tremendous drop in explanation quality (AUC-ROC). However, BottleneckMLP acts as such entropy regularizer, and the AUC-ROC goes right back up, reaching or exceeding the initial PGExplainer performance.

Table 17: BottleneckMLP improves PGExplainer Explanation AUC-ROC for BA-Shapes and Tree-Grids. For TreeCycles, BottleneckMLP recovers the function of the Entropy Loss and achieved almost equivalent performance to the baseline.

|  | BA-Shapes | Tree-Cycles | Tree-Grids |
|---|---|---|---|
| PGExplainer | $\underline{0.993} \pm 0.006$ | $\mathbf{0.941} \pm 0.002$ | $\underline{0.676} \pm 0.003$ |
| PGExplainer w/o Entropy Loss | $0.033 \pm 0.021$ | $0.058 \pm 0.002$ | $0.628 \pm 0.028$ |
| PGExplainer w/o E. Loss + BottleneckMLP | $\mathbf{0.999} \pm 0.0001$ | $\underline{0.938} \pm 0.031$ | $\mathbf{0.732} \pm 0.001$ |

### H.3    BottleneckMLP for Post-hoc Graph Classification

#### H.3.1    V-InFoR

V-InFoR Wang et al. (2023) approaches GNN explanation by first denoising corrupted graphs using a variational latent distribution of graph representations, then generating explanations by graph information bottleneck (GIB). We remove this GIB term and replace it with our BottleneckMLP. Table 18 demonstrates the superior performance of V-InFoR when we add BottleneckMLP on Fidelity, Sufficiency, Necessity, $F_{ns}$ Score, and Mean Probability. BottleneckMLP increases performance on all metrics, with the exception for Prob-N for BA3, where the value is only $0.003$ lower, and for Mean Probability on MUTAG.

Table 18: Comparison of Fidelity and Explanation Metrics for BA3 and MUTAG on V-InFoR. Adding BottleneckMLP to V-InFoR increases performance on all five metrics on both datasets, with the exception of a slight decrease in Prob-N for BA3, and a decrease in Mean Probability with MUTAG.

| Dataset | Metric | V-InFoR | V-InFoR w/o Info Loss | V-InFoR w/o Info Loss + BottleneckMLP |
|---------|--------|---------|-----------------------|----------------------------------------|
| BA3 | Fidelity | $\underline{0.5240} \pm 0.0096$ | $0.5192 \pm 0.0129$ | $\mathbf{0.5308} \pm 0.0154$ |
| | Prob-S | $0.5105 \pm 0.0123$ | $\underline{0.5218} \pm 0.0156$ | $\mathbf{0.5236} \pm 0.0199$ |
| | Prob-N | $\mathbf{0.6620} \pm 0.0010$ | $0.6595 \pm 0.0035$ | $\underline{0.6617} \pm 0.0026$ |
| | $F_{ns}$ Score | $0.5764 \pm 0.0073$ | $\underline{0.5825} \pm 0.0103$ | $\mathbf{0.5844} \pm 0.0120$ |
| | Mean Probability | $\underline{0.6982} \pm 0.1981$ | $0.5104 \pm 0.1091$ | $\mathbf{0.7607} \pm 0.1860$ |
| MUTAG | Fidelity | $\underline{0.5718} \pm 0.0098$ | $0.5240 \pm 0.0096$ | $\mathbf{0.5992} \pm 0.0159$ |
| | Prob-S | $\underline{0.5718} \pm 0.0098$ | $0.5105 \pm 0.0123$ | $\mathbf{0.5992} \pm 0.0159$ |
| | Prob-N | $\underline{0.4749} \pm 0.0122$ | $0.4488 \pm 0.0182$ | $\mathbf{0.6620} \pm 0.0010$ |
| | $F_{ns}$ Score | $\underline{0.5187} \pm 0.0088$ | $0.5130 \pm 0.0148$ | $\mathbf{0.5764} \pm 0.0073$ |
| | Mean Probability | $\mathbf{0.9675} \pm 0.0053$ | $0.4809 \pm 0.1925$ | $\underline{0.6982} \pm 0.1981$ |

#### H.3.2    ReFine

ReFine Wang et al. (2021) introduces a two-stage framework for graph explanation that combines global class-level reasoning with graph-specific refinement. In the pre-training stage, ReFine learns to distinguish graphs of different classes by optimizing a contrastive objective, enabling it to capture high-level, class-discriminative structural patterns. In the fine-tuning stage, these global cues are adapted to each individual graph, producing explanations that account for its local structure.

Table 19 shows that ReFine with BottleneckMLP has comparable performance to the baseline ReFine model on BA3 and MNIST datasets. BottleneckMLP improves ACC-AUC on MUTAG by $19.9\%$.

Table 19: Performance comparison (ACC-AUC and Mean Precision) across BA3, MNIST, and MUTAG for ReFine variants. BottleneckMLP improves performance most significantly on MUTAG.

| Model Variant | BA3 | MNIST | MUTAG |
|---------------|-----|-------|-------|
| ReFine | $\mathbf{0.559}$ | $\underline{0.261}$ | $\underline{0.694}$ |
| ReFine w/o fidelity loss | $0.505$ | $0.199$ | $0.664$ |
| ReFine w/ BottleneckMLP | $\underline{0.552}$ | $\mathbf{0.263}$ | $\mathbf{0.832}$ |

### H.4    ProxyExplainer

ProxyExplainer Chen et al. (2024) is a recent post-hoc method that leverages 'proxy graphs' generated by a Graph VAE to generate in-distribution explanations. Table 20 demonstrates the performance increase in Explanation AUC-ROC with the addition of BottleneckMLP.

Table 20: ProxyGraph Explanatation AUC and Fidelity+/-

| Model | Dataset | AUC | Fidelity+ | Fidelity- |
|-------|---------|-----|-----------|-----------|
| **ProxyExplainer** | Fluoride Carbonyl | $0.518 \pm 0.081$ | $0.050 \pm 0.021$ | $\mathbf{0.409} \pm 0.066$ |
| | Alkane Carbonyl | $0.211 \pm 0.075$ | $0.112 \pm 0.039$ | $0.503 \pm 0.184$ |
| | BA-2Motifs | $\mathbf{0.940} \pm 0.011$ | $\mathbf{0.509} \pm 0.015$ | $0.201 \pm 0.006$ |
| | MUTAG | $0.847 \pm 0.139$ | $0.714 \pm 0.049$ | $0.345 \pm 0.127$ |
| **ProxyExplainer + BottleneckMLP** | Fluoride Carbonyl | $\mathbf{0.681} \pm 0.026$ | $\mathbf{0.102} \pm 0.029$ | $0.633 \pm 0.032$ |
| | Alkane Carbonyl | $\mathbf{0.527} \pm 0.332$ | $\mathbf{0.775} \pm 0.142$ | $\mathbf{0.253} \pm 0.176$ |
| | BA-2Motifs | $0.779 \pm 0.032$ | $0.500 \pm 0.011$ | $\mathbf{0.000} \pm 0.000$ |
| | MUTAG | - | - | - |

## H.5 BOTTLENECKMLP FOR NON-IB-BASED ANTE-HOC GRAPH CLASSIFICATION

### H.5.1 GOAT

GOAt Lu et al. (2024) learns stochastic, edge-level gates that selectively preserve only the most task-relevant subgraph during message passing, producing graph explanations directly from the forward pass. Table 21 demonstrates the superior performance of GOAt when we add BottleneckMLP on explanation quality, measured by Fidelity-, Fidelity+, and Sparsity. BottleneckMLP increases performance on all metrics, with the exception for the GOAt-GCN variant on BA-2Motifs. All other datasets and configurations have increased performance with BottleneckMLP.

Table 21: Explanation Quality Metrics (Fidelity-, Fidelity+, Sparsity) across GOAt Variants and Datasets. BottleneckMLP increases performance.

| Model | Dataset | Fidelity- | Fidelity+ | Sparsity |
|-------|---------|-----------|-----------|----------|
| GOAt-GCN | BA-2Motifs | $\mathbf{0.001} \pm 0.048$ | $\mathbf{0.542} \pm 0.012$ | $0.786 \pm 0.076$ |
| | Mutagenicity | $0.140 \pm 0.040$ | $0.653 \pm 0.047$ | $0.764 \pm 0.013$ |
| | NCI1 | $0.086 \pm 0.003$ | $0.489 \pm 0.072$ | $0.835 \pm 0.000$ |
| GOAt-GCN + BottleneckMLP | BA-2Motifs | $0.056 \pm 0.047$ | $0.509 \pm 0.005$ | $\mathbf{0.788} \pm 0.041$ |
| | Mutagenicity | $\mathbf{0.073} \pm 0.196$ | $\mathbf{0.702} \pm 0.076$ | $\mathbf{0.786} \pm 0.021$ |
| | NCI1 | $\mathbf{0.004} \pm 0.142$ | $\mathbf{0.563} \pm 0.058$ | $\mathbf{0.848} \pm 0.018$ |
| GOAt-GIN | BA-2Motifs | $\mathbf{0.001} \pm 0.000$ | $0.573 \pm 0.004$ | $0.836 \pm 0.001$ |
| | Mutagenicity | $0.436 \pm 0.003$ | $0.650 \pm 0.003$ | $0.808 \pm 0.000$ |
| | NCI1 | $0.590 \pm 0.110$ | $0.744 \pm 0.021$ | $0.799 \pm 0.001$ |
| GOAt-GIN + BottleneckMLP | BA-2Motifs | $0.070 \pm 0.000$ | $\mathbf{0.977} \pm 0.001$ | $\mathbf{0.845} \pm 0.002$ |
| | Mutagenicity | $\mathbf{0.020} \pm 0.002$ | $\mathbf{0.692} \pm 0.015$ | $\mathbf{0.843} \pm 0.000$ |
| | NCI1 | $\mathbf{0.500} \pm 0.014$ | $\mathbf{0.783} \pm 0.001$ | $\mathbf{0.810} \pm 0.002$ |
| GOAt-SAGE | BA-2Motifs | $0.479 \pm 0.004$ | $0.500 \pm 0.004$ | $0.813 \pm 0.014$ |
| | Mutagenicity | $0.040 \pm 0.018$ | $0.525 \pm 0.005$ | $0.812 \pm 0.009$ |
| | NCI1 | $0.506 \pm 0.174$ | $0.674 \pm 0.051$ | $0.803 \pm 0.013$ |
| GOAt-SAGE + BottleneckMLP | BA-2Motifs | $\mathbf{0.472} \pm 0.004$ | $\mathbf{0.500} \pm 0.001$ | $\mathbf{0.833} \pm 0.001$ |
| | Mutagenicity | $\mathbf{0.013} \pm 0.021$ | $\mathbf{0.572} \pm 0.003$ | $\mathbf{0.823} \pm 0.001$ |
| | NCI1 | $\mathbf{0.391} \pm 0.001$ | $\mathbf{0.728} \pm 0.001$ | $\mathbf{0.808} \pm 0.012$ |

### H.5.2 DIR

DIR (Wu et al., 2022) aims to learn causal rationales and confounding rationales to improve explanation. The method introduces an invariance loss which we consider as the "Info Loss" term to ablate in their method, as it is the analogous component of IB loss in DIR. Table **??** demonstrates the performance of DIR when we add BottleneckMLP on Causal ACC, Confounding ACC, Train ACC, Validation ACC, Test Precision@5, and Test MRR. BottleneckMLP increases Train ACC, Validation ACC, and Confounding ACC. In other most cases, DIR with BottleneckMLP slightly under-performs DIR, however, BottleneckMLP is able to recover the performance increase relatedba to DIR when we remove the Info Loss terms, showing that it implicitly causes the same effects as does the explicit loss in DIR.

Table 22: DIR results comparing Original vs. BottleneckMLP across datasets. BottleneckMLP consistently improves Train ACC, Val ACC, and Conf ACC. For Causal ACC, Test Prec, and Test MRR, results are similar, and BottleneckMLP replaces the full function of the Info Loss in DIR.

| Dataset | Method | Causal ACC | Conf ACC | Train ACC | Val ACC | Test Prec | Test MRR |
|---|---|---|---|---|---|---|---|
| spmotif-0.9 | DIR | **0.3921**±0.0607 | 0.3548±0.0216 | 0.7303±0.0974 | 0.4920±0.0963 | **0.2251**±0.0928 | **0.3828**±0.1394 |
| | DIR w/o Info Loss | 0.3488±0.0181 | 0.3386±0.0058 | 0.8780±0.0262 | 0.4167±0.1089 | 0.1499±0.1234 | 0.2607±0.2175 |
| | BottleneckMLP | 0.3505±0.0247 | **0.3376**±0.0098 | **0.9069**±0.0411 | **0.4928**±0.1687 | 0.1927±0.1432 | 0.3434±0.2640 |
| spmotif-0.7 | DIR | **0.4089**±0.0570 | 0.3669±0.0079 | 0.6916±0.1210 | 0.5592±0.1478 | **0.2013**±0.1082 | **0.3521**±0.1669 |
| | DIR w/o Info Loss | 0.3708±0.0400 | 0.3582±0.0370 | 0.7291±0.0450 | 0.4888±0.1376 | 0.1475±0.1204 | 0.2556±0.2103 |
| | DIR w/ BottleneckMLP | 0.3794±0.0389 | **0.3377**±0.0030 | **0.8137**±0.1123 | **0.6048**±0.2355 | 0.1856±0.1292 | 0.3358±0.2392 |
| spmotif-0.5 | DIR | **0.3918**±0.0444 | 0.3388±0.0100 | 0.6696±0.0661 | 0.6064±0.0935 | **0.2227**±0.1104 | **0.3842**±0.1566 |
| | DIR w/o Info Loss | 0.3590±0.0406 | 0.3419±0.0101 | 0.6280±0.2131 | 0.5347±0.2548 | 0.1576±0.1341 | 0.2848±0.1987 |
| | DIR w/ BottleneckMLP | 0.3907±0.0234 | **0.3363**±0.0037 | **0.7256**±0.0899 | **0.6417**±0.1320 | 0.1908±0.1270 | 0.3534±0.2316 |
| MNIST-75sp | DIR | 0.2112±0.0297 | 0.1540±0.0209 | 0.2643±0.1018 | 0.2665±0.1018 | – | – |
| | DIR w/o Info Loss | 0.1572±0.0348 | 0.1655±0.0092 | 0.2600±0.0949 | 0.2600±0.0949 | – | – |
| | DIR w/ BottleneckMLP | **0.2130**±0.0271 | **0.1529**±0.0276 | **0.3736**±0.1794 | **0.3621**±0.1809 | – | – |
| Graph-SST2 | DIR | **0.8200**±0.0099 | 0.8169±0.0096 | **0.9949**±0.0024 | **0.9248**±0.0018 | – | – |
| | DIR w/o Info Loss | 0.8001±0.0087 | 0.8093±0.0095 | 0.9885±0.0096 | 0.9238±0.0095 | – | – |
| | DIR w/ BottleneckMLP | 0.8101±0.0174 | **0.8000**±0.0178 | 0.9915±0.0055 | 0.9244±0.0053 | – | – |

## H.6 BOTTLENECKMLP FOR TEMPORAL GRAPH CLASSIFICATION

Table 23: TGIB Classifier Accuracy.

| | **Classifier Accuracy** | | |
|---|---|---|---|
| **Model** | Wikipedia | CanParl | USLegis |
| TGIB | 0.947 | 0.528 | **0.642** |
| TGIB w/o Info Loss | **0.960** | **0.588** | 0.544 |
| TGIB w/o Info Loss + Bottleneck MLP | 0.959 | 0.586 | 0.607 |

## I EXPLICIT IB METHODS

We demonstrate ineffectiveness across multiple methods:

**GSAT** Miao et al. (2022) adopts a variational IB framework to extract explanatory subgraphs. It introduces a lower bound on $I(G_S; Y)$ by employing a variational approximation to the joint distribution $P(Y, G_S)$, and an upper bound on $I(G; G_S)$ (what we refer to as info-loss) using a variational approximation to the marginal $P(G_S) = \sum_G P_\phi(G_S|G)P_G(G)$.

The resulting KL-divergence term between $P_\phi(G_S|G)$ and $Q(G_S)$ simplifies to a sum of KL divergences between individual Bernoulli distributions per edge. While this formulation is differentiable, it is not tight. The prior $Q(G_S)$ acts only as a weak regularizer since it assumes i.i.d. edge inclusion and ignores structural dependencies. Consequently, the KL term provides an upper bound on $I(G; G_S)$ which is unable to sufficiently constrain the learned explainer.

**PGIB** Seo et al. (2024b) introduces a prototype-based framework that explains GNN predictions by selecting a subgraph $G_s$ and a prototype $G_p$ that together retain information about the label $Y$. It maximizes a lower bound on $I(Y; G_s, G_p)$ via a variational classifier $q_\theta(Y|\gamma(G_s, G_p))$ where $\gamma$ is a similarity function. To minimize $I(G_s; G_p)$ (what we refer to as info-loss), PGIB uses a variational upper bound $\mathbb{E}[-\log q_\phi(G_p|G_s)]$ similar to GSAT, or the variant of their method, PGIB$_{\text{CONT}}$, leverages a contrastive loss approach proposed in Rusak et al. (2025) to minimize $I(X, Z)$, see Appendix M.

Despite being grounded in the IB framework, these approximations are weak. The variational classifier $q_\theta$ is task-specific and does not tightly control information flow similar to GSAT. The contrastive loss in PGIB$_{\text{CONT}}$ is indirect, sensitive to sampling and hyperparameters, and lacks clear control over mutual information. As such, PGIB$_{\text{CONT}}$ does not impose a tight constraint on $I(G; G_S)$, as it does not explicitly regularize the information retained from the original graph $G$ and enforce compression.

**TGIB** integrates temporal graph learning with the IB principle to improve both link prediction and model explainability. It extracts a bottleneck code $R_k$ for each target edge $e_k$ from its $L$-hop neighborhood $G_k$. This code is a subgraph of the $L$-hop computation graph around $e_k$ and acts as a compressed representation used for prediction. By limiting the information flow through this bottleneck, the model highlights the relevant parts of the neighborhood for edge prediction.

The objective function is defined as: $\min_{R_k} -I(Y_k; R_k) + \beta I(R_k; e_k, G_k)$ where the first term encourages the model to preserve information relevant to the label $Y_k$, while the second term penalizes encoding unnecessary information from the edge $e_k$ and its surrounding graph $G_k$. We find that the IB Loss is detrimental to the performance of TGIB.

## J  FIDELITY METRIC

Following Seo et al. (2024b), we use Fidelity metrics Pope et al. (2019); Yuan et al. (2023) to quantify the quality of the explanations. Let $y_i$ denote the ground-truth values and $\hat{y}_i$ denote the predicted values for the $i$-th input graph. Let $k$ be the sparsity score, denoting the $(k \times 100)\%$ of important nodes of the original graph which are used to construct the explanatory subgraph. The prediction of the explanatory subgraph for a sparsity score of $k$ is denoted by $\hat{y}_i^k$. The prediction of the non-explanatory subgraph for a sparsity score of $k$ is denoted by $\hat{y}^{1-k_i}$. The equations for the Fidelity metrics are given by:

$$F^- = \frac{1}{N}\sum_{i=1}^{N} \mathbb{I}(y_i = \hat{y}_i) - \mathbb{I}(y_i = \hat{y}_i^k), F^+ = \frac{1}{N}\sum_{i=1}^{N} \mathbb{I}(y_i = \hat{y}_i) - \mathbb{I}(y_i = \hat{y}_i^{1-k}), \quad (12)$$

where the binary indicator $\mathbb{I}(y_i = \hat{y}_i)$ returns 1 if $y_i = \hat{y}_i$, and 0, otherwise. Higher values for $F^+$, and lower values for $F^-$ indicate that the explanatory subgraphs produced by the model are better.

## K  NORMALIZED SPACE ALIGNMENT (NSA)

To evaluate the similarity between learned representations in neural networks, Ebadulla et al. propose Normalized Space Alignment (NSA) as a manifold analysis technique for neural network representations which provides a robust similarity metric, and loss function, for comparing vector spaces across architectures, layers, or training regimes Ebadulla et al. (2025). NSA builds upon previous methods including Canonical Correlation Analysis (CCA) Morcos et al. (2018); Raghu et al. (2017) and Centered Kernel Alignment (CKA) Kornblith et al. (2019), but addresses key limitations such as scale sensitivity and confounding dimensionality effects.

The central idea is to treat representations as subspaces, and then measure alignment via projections onto the normalized Grassmann manifold, where vector directions are invariant to orthogonal transformations and global scaling. Ebadulla et al. introduce Global NSA (GNSA), which compares the entire representation spaces holistically, based on normalized projection matrices and Frobenius norms, and Local NSA (LNSA), which focuses on fine-grained, per-sample neighborhood structure preservation via rank-correlated similarity matrices, emphasizing local geometric alignment. NSA provides a scale- and dimension-agnostic framework to evaluate whether our BottleneckMLP layers yield more robust and aligned task-relevant representations.

## L  DERIVATIONS

$$I(X; Z) = \mathbb{E}_{X,Z}\left[\log \frac{p(x \mid z)}{p(x)}\right]$$

$$= \mathbb{E}_{X,Z}\left[\log \prod_i \frac{p(x_i \mid \text{Patch}(x_i), z)}{p(x_i \mid \text{Patch}(x_i))}\right] = \mathbb{E}_{X,Z}\left[\sum_i \log \frac{p(x_i \mid \text{Patch}(x_i), z)}{p(x_i \mid \text{Patch}(x_i))}\right] \quad (13)$$

# M  PGIB

The aforementioned contrastive loss approach used in PGIB is given as follows:

$$\mathcal{L}_{\text{CONT}} := -\frac{1}{n}\sum_{i=1}^{n}\log\frac{\exp(g(z_{G_i}, z_{G_j})/\tau)}{\sum_{k:z_k\notin P_{\text{sub}}}\exp(g(z_{G_i}, z_{G_k})/\tau)}. \tag{14}$$

# N  RUNNING TIME ANALYSIS OF BOTTLENECKMLP

Adding BottleneckMLP has no effect on the running time of baseline models.

Table 24: GSAT Training Mode Runtime (in minutes)

| Mode | MUTAG | BA-2Motifs | NCI1 | PROTEINS |
|---|---|---|---|---|
| GSAT | $8.25 \pm 0.70$ | $4.03 \pm 0.33$ | $10.76 \pm 0.96$ | $7.23 \pm 0.91$ |
| GSAT w/o IB Loss | $9.15 \pm 0.64$ | $4.04 \pm 0.18$ | $10.26 \pm 0.39$ | $6.82 \pm 0.95$ |
| GSAT w/o IB Loss + BottleneckMLP | $9.47 \pm 0.97$ | $4.18 \pm 0.15$ | $10.66 \pm 0.64$ | $7.10 \pm 1.04$ |

Table 25: PGIB Training Runtime (in minutes)

| Mode | MUTAG | BA-2Motifs | NCI1 | PROTEINS |
|---|---|---|---|---|
| PGIB (Normal) | $13.01 \pm 1.36$ | $5.46 \pm 0.17$ | $9.87 \pm 1.38$ | $91.78 \pm 7.88$ |
| PGIB w/o IB Loss | $15.90 \pm 3.37$ | $5.13 \pm 0.28$ | $9.90 \pm 1.00$ | $69.17 \pm 11.97$ |
| PGIB w/o IB Loss + BottleneckMLP | $9.95 \pm 3.01$ | $5.36 \pm 0.74$ | $12.75 \pm 0.84$ | $86.96 \pm 14.07$ |

# O  CONVERGENCE ANALYSIS

PGIB with BottleneckMLP converges fastest, and most consistently as compared with original PGIB, and PGIB without InfoLoss. BottleneckMLP effects the node embeddings by injecting noise scaled inversely by node importance, making important node embeddings more stable, and causing unimportant node embeddings to drift. Hence, this accelerates the representation dynamics of the node embeddings over training epochs, leading to lower epochs needed for model convergence (Table 26), and improved performance.

With $K$ hidden layers forming a Markov chain $X \to Z_1 \to \cdots \to Z_K$, each layer performs a partial compression, $\Delta I_X \approx \sum_k \Delta I_X^k$. The time required to achieve compression of magnitude $\Delta I_X$ via diffusion scales exponentially (Equation 15) as $\exp(\frac{\Delta I_X}{D})$, where $D$ is a diffusion constant.

$$\exp\left(\sum_k \Delta I_X^k\right) \gg \sum_k \exp\left(\Delta I_X^k\right) \tag{15}$$

Hence, convergence accelerates exponentially. As such, our BottleneckMLP reaches the compression phase significantly faster and achieves more effective compression than explicit IB regularization.

# P  COMPARATIVE ANALYSIS OF INFORMATION-PLANE DYNAMICS ACROSS MODELS

To understand how architectural choices influence information flow in graph neural networks, we compare the information-plane trajectories of three model variants. For each model, we plot $(I(X; Z_i))$ versus $(I(Z_i; Y))$ across training epochs (color-coded from purple to yellow). We show

Table 26: Comparison of convergence epoch for original PGIB, PGIB w/o IB Loss, and PGIB w/o IB Loss + BottleneckMLP configurations across datasets. All results are averaged over 10 seeds. The maximum number of epochs was 500.

| Dataset/ Mode | PGIB | PGIB w/o IB Loss | PGIB w/o IB Loss + BottleneckMLP |
|---|---|---|---|
| BA 2Motifs | $239 \pm 136$ | $256 \pm 167$ | $\mathbf{149} \pm 54$ |
| MUTAG | $341 \pm 134$ | $412 \pm 112$ | $\mathbf{240} \pm 65$ |
| NCI1 | $216 \pm 148$ | $219 \pm 162$ | $\mathbf{160} \pm 172$ |
| PROTEINS | $500 \pm 0$ | $500 \pm 0$ | $\mathbf{358} \pm 174$ |
| FluorideCarbonyl | $396 \pm 136$ | $250 \pm 177$ | $\mathbf{246} \pm 55$ |
| AlkaneCarbonyl | $397 \pm 155$ | $246 \pm 135$ | $\mathbf{250} \pm 77$ |
| Benzene | $346 \pm 189$ | $260 \pm 189$ | $\mathbf{192} \pm 156$ |

that the characteristic IB curve does not arise naturally in GNNs, but does arise with the addition of architectural constraints via BottleneckMLP.

**Baseline Model.** Figure 13a and 14a shows that the baseline GNN does not exhibit the expected IB behavior. Across both layers, the scatter of points remains diffuse with no clear epoch-wise trajectory. The mutual information with the input, $(I(X; Z_i))$, remains relatively high throughout training, while $(I(Z_i; Y))$ does not show late-stage concentration. Early and late epochs are mixed together, indicating little systematic change in information content over training. These results confirm that, in standard GNN architectures, compression does not naturally emerge even with cross-entropy optimization.

**Baseline Model w/o Info Loss.** When removing the explicit information loss term (Figure 13b and 14b), the model retains even more information about the inputs, reflected by substantially higher $(I(X; Z_i))$ values compared to the baseline. While there is a weak upward trend in $(I(Z_i; Y))$ for later epochs, there is still no evidence of compression in $(I(X; Z_i))$.

**BottleneckMLP.** In contrast, the with BottleneckMLP, (Figure 3) displays a pronounced IB structure. Early layers behave similarly to those in the baseline model, but beginning around Layer 3, clear compression in $(I(X; Z_i))$ emerges as training progresses. Points transition from right (high $(I(X; Z_i))$) to left (compressed representations) while $(I(Z_i; Y))$ becomes more concentrated. By Layers 4 and 5, the classical IB trajectory is clearly visible: the model discards redundant input information while retaining label-relevant structure. This trend aligns with theoretical expectations and with our discussion in Section 4.3.1.

Collectively, these observations demonstrate that GNNs do not naturally follow IB dynamics, even under standard cross-entropy optimization. The IB curve only emerges when compression is architecturally enforced through an explicit bottleneck.

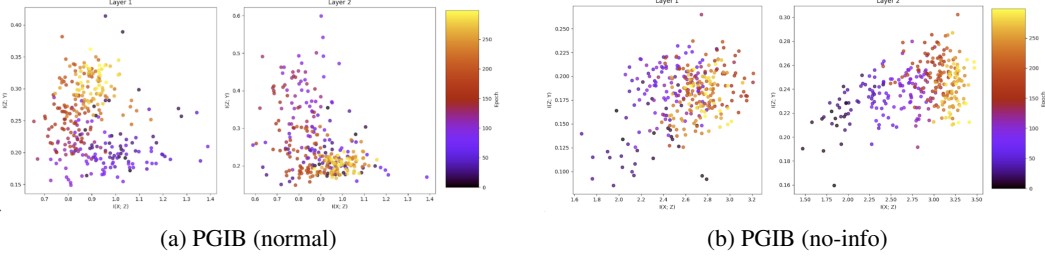

(a) PGIB (normal)  (b) PGIB (no-info)

Figure 13: PGIB information plane $I(X; Z_i)$ vs. $I(Z_i; Y)$ over baseline GNN layers with (left) and without(right) Info Loss, across epochs (purple to yellow). Information curve Tishby & Zaslavsky (2015) does not appear.

## Q  PSEUDOCODE

We provide the pseudocode for the importance-weighted noise injection of BottleneckMLP during the training of each explainer model in Algorithm 1.

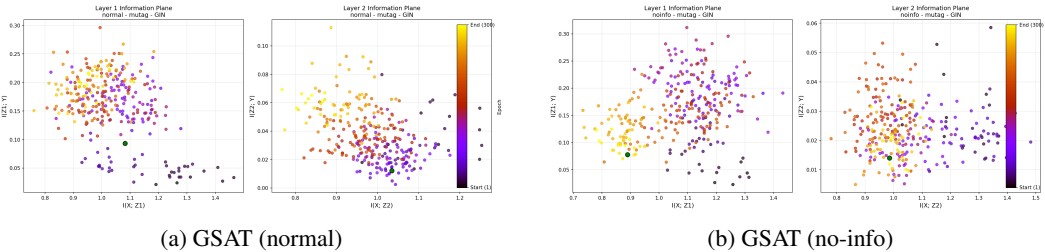

(a) GSAT (normal)            (b) GSAT (no-info)

Figure 14: GSAT information plane $I(X; Z_i)$ vs. $I(Z_i; Y)$ over baseline GNN layers with (left) and without(right) Info Loss, across epochs (purple to yellow). Information curve Tishby & Zaslavsky (2015) does not appear.

---

**Algorithm 1:** Importance-Weighted Noise Injection Training Loop

---
**Input:** Graph $G$, Label $y$
**Input:** Classifier $f_\theta$, subgraph extractor $g_\phi$, noise scale $\sigma$
Initialize parameters $\phi, \theta$
**for** $epoch \leftarrow 1$ **to** $E$ **do**
 $Z \leftarrow f(G)$ ;       // Initial node embeddings
 $\alpha \leftarrow \sigma(g_\phi(Z))$ ;    // Importance score generation
 Sample $\delta \sim \mathcal{N}(0, I_d)$
 $S \leftarrow \dfrac{\sigma}{\sqrt{\alpha}}$
 $\tilde{Z} \leftarrow Z_B + S \odot \delta$ ;   // Importance-weighted noise
 $\hat{Y} \leftarrow h_\psi(\tilde{Z}, \alpha)$
 $\mathcal{L} \leftarrow \text{CrossEntropy}(Y_B, \hat{Y})$ ;
 Update $\phi, \theta$ to minimize $\mathcal{L}$

---

## R  GNN BACKBONES

We ran additional experiments with various GNN backbones to demonstrate the general applicability of BottleneckMLP. These results are shown for test accuracy and explanation AUROC for GSAT.

Table 27: Accuracy and AuROC results for different backbone models and BottleneckMLP variants.

| Model Variant | MUTAG (Acc) | NCI1 (Acc) | PROTEINS (Acc) | MUTAG (AuROC) |
|---|---|---|---|---|
| GAT (normal) | $0.893 \pm 0.026$ | $0.726 \pm 0.016$ | $\mathbf{0.893 \pm 0.026}$ | $\mathbf{0.690 \pm 0.304}$ |
| GAT (noinfo) | $0.892 \pm 0.025$ | $0.727 \pm 0.015$ | $\underline{0.892 \pm 0.025}$ | $0.431 \pm 0.301$ |
| GAT + BottleneckMLP (64–48–32) | $\mathbf{0.913 \pm 0.021}$ | $\mathbf{0.748 \pm 0.026}$ | $0.725 \pm 0.046$ | $\underline{0.482 \pm 0.390}$ |
| GAT + BottleneckMLP (64–16) | $\underline{0.900 \pm 0.020}$ | $\underline{0.743 \pm 0.025}$ | $0.733 \pm 0.041$ | $0.478 \pm 0.317$ |
| GCN (normal) | $0.885 \pm 0.027$ | $0.726 \pm 0.014$ | $\underline{0.713 \pm 0.048}$ | $0.534 \pm 0.369$ |
| GCN (noinfo) | $0.893 \pm 0.033$ | $\underline{0.727 \pm 0.015}$ | $0.702 \pm 0.047$ | $\underline{0.571 \pm 0.346}$ |
| GCN + BottleneckMLP (64–48–32) | $\underline{0.908 \pm 0.027}$ | $\mathbf{0.743 \pm 0.013}$ | $\mathbf{0.746 \pm 0.043}$ | $0.520 \pm 0.403$ |
| GCN + BottleneckMLP (64–16) | $\mathbf{0.915 \pm 0.021}$ | $\mathbf{0.743 \pm 0.013}$ | $\mathbf{0.746 \pm 0.043}$ | $\mathbf{0.836 \pm 0.129}$ |
| PNA (normal) | $\underline{0.939 \pm 0.029}$ | $0.734 \pm 0.026$ | $0.640 \pm 0.061$ | $\mathbf{0.993 \pm 0.005}$ |
| PNA (noinfo) | $\mathbf{0.947 \pm 0.017}$ | $\mathbf{0.781 \pm 0.028}$ | $0.705 \pm 0.093$ | $\underline{0.972 \pm 0.008}$ |
| PNA + BottleneckMLP (64–16) | $\underline{0.939 \pm 0.015}$ | $0.749 \pm 0.026$ | $\mathbf{0.725 \pm 0.052}$ | $0.965 \pm 0.010$ |
| PNA + BottleneckMLP (64–48–32) | $0.935 \pm 0.021$ | $\underline{0.760 \pm 0.025}$ | $0.695 \pm 0.069$ | $0.959 \pm 0.012$ |

## S  HYPERPARAMETER ANALYSIS

We include analysis of the hyperparameter $\sigma$ which parametrizes the maximum gaussian noise variance in our method. We tune this for different datasets across different methods (GSAT, PGIB...).

Figure 15 shows that GSAT classifier accuracy is not too sensitive to different $\sigma$ values, and we find the sweet spot for accuracy with grid search. Explanation AUC-ROC drops for too high values of $\sigma$ in the early stages of training, which detriments learning.

Figure 15: GSAT Classifier Accuracy for Different Sigma Values

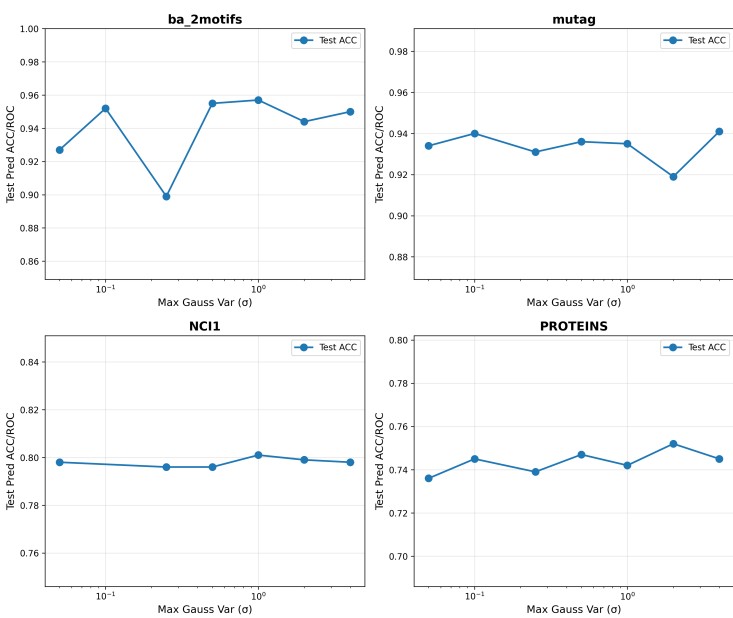

Figures 17 and 18 show the sensitivity analysis for $\sigma$ on PGIB with datasets MUTAG and BA-2Motifs. The optimal value of $\sigma$ varies across datasets and methods. $\sigma$ increases classifier and explanation performance, however, too high values of $\sigma$ will introduce too much noise during early training, as reflected in the performance drop at higher values of $\sigma$.

## T  SPARSITY

We use the default top-k protocol followed by each baseline method. That is, for each method, we retain the same k used by the original explainer (e.g., k = 5 or k = 10 depending on the dataset/benchmark). This ensures that BottleneckMLP is evaluated under the same sparsity budget as the original method, making the comparison fair and directly aligned with prior work.

While standard IB adaptively balances compression and prediction via the IB trade-off parameter ($\beta$), BottleneckMLP does not include an explicit sparsity-controlling term. Instead, sparsity emerges implicitly from the entropy separation induced by the noise injection: important nodes stabilize in

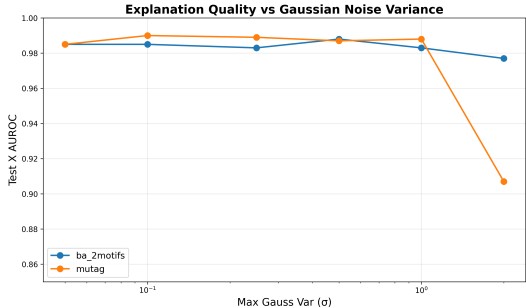

Figure 16: GSAT Explanation AUC-ROC for Different Sigma Values

Figure 17: PGIB Classifier Accuracy for Different Sigma Values

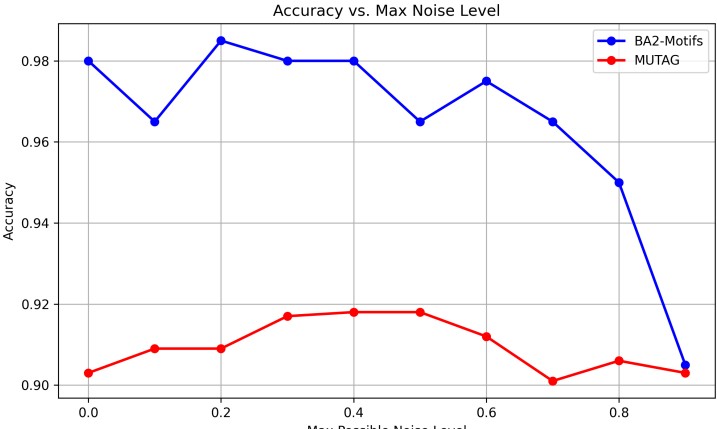

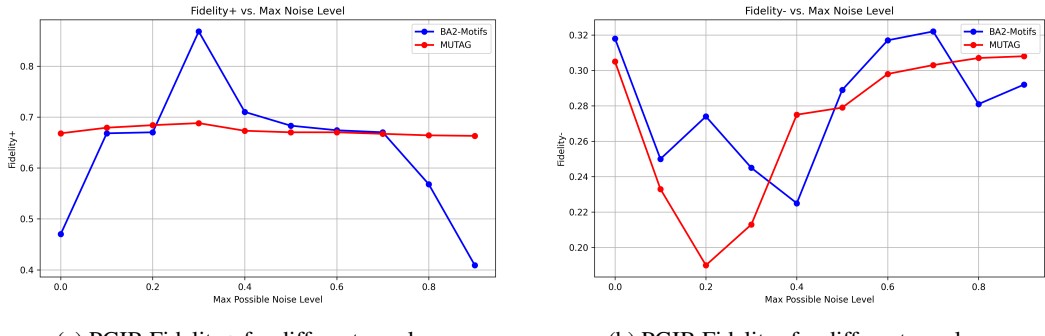

(a) PGIB Fidelity+ for different $\sigma$ values        (b) PGIB Fidelity- for different $\sigma$ values

Figure 18: PGIB Fidelity+ and Fidelity- as a function of noise level $\sigma$.

low-entropy clusters, while unimportant nodes drift toward Gaussianized, high-entropy distributions.

In Figure 19, we show the effect of differing sparsity values, k, where k is the percent of edges in the explanatory subgraph.

Figure 19: PGIB Fidelity+/ Fidelity- for Different Sparsity Values

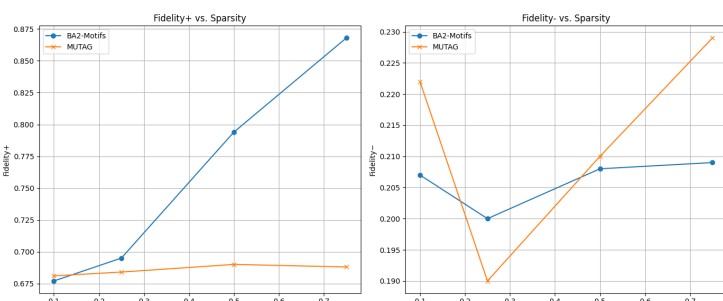

