# OpenReview forum: "BottleneckMLP: Graph Explanation via Implicit Information Bottleneck"
_ICLR.cc/2026/Conference — Submitted to ICLR 2026_

### Official Review · Reviewer_Bvh8 · 2025-10-16

**Soundness:** 2
**Presentation:** 3
**Contribution:** 2
**Rating:** 2
**Confidence:** 4

**Summary:**

This paper proposes a new architectural approach to improve the interpretability of Graph Neural Networks (GNNs) through implicit enforcement of the Information Bottleneck (IB) principle, without relying on explicit IB loss terms. Although the paper’s title sounds broad, the actual problem scope is much narrower, where they only focus on improving Ante-hoc and IB-based GNNExplainablity. Their goal is to show that explicit IB is unnecessary even within IB-based ante-hoc frameworks.

**Strengths:**

1. Their experiments demonstrate that adding BottleneckMLP often outperforms using explicit IB loss terms in existing ante-hoc explainers (GSAT, PGIB, TGIB) across multiple datasets. The results are effective.
2. They provide analysis showing how unimportant node embeddings are pushed toward Gaussian, high-entropy distributions (i.e. “forgetting”), while important nodes remain structured, thereby aligning with the IB principle in a principled manner.

**Weaknesses:**

1. The claim that “a prominent line of work leverages the IB principle” is overclaim. IB-based explainers represent only a small subset of existing methods, and most state-of-the-art explainers (e.g., SubgraphX, GOAt, ReFine, PGExplainer++) do not rely on IB and typically perform better.
2. By restricting comparisons only to IB-related baselines (GSAT, PGIB, TGIB) and including only one outdated non-IB baseline (PGExplainer, 2020), the experimental validation becomes narrow and unconvincing. Broader comparison with recent non-IB explainers is necessary to justify the contribution.
3. Explaining temporal GNNs is within their scope, but the evaluation is incomplete. They do not clearly justify the pros and cons of applying IB to TGNNs, and they fail to compare with known temporal explainers like T-GNNExplainer.
4. They compare only to IB-based explainers, but they should also include other ante-hoc GNN explainers that are not IB-based.

**Questions:**

Why limit baseline to GCN? Why not test on GIN, which is stronger than GCN in many graph tasks? Their experimental setup seems biased by weak baselines.

---

> ### Author Response · Authors · 2025-11-21
> **Response to Reviewer Bvh8**
>
> We thank the reviewer for the constructive feedback.
>
> (W1&2) We agree that IB-based explainers represent only one subset of the broader GNN explainability landscape, and that many state-of-the-art methods (e.g., GOAt, ReFine, V-InFoR, DIR) are not IB-based. We have revised the introductory phrasing to: ``a growing line of work leverages the Information Bottleneck (IB) principle for GNN explanation.’’ We now explicitly cite representative ante-hoc and post-hoc IB methods (GSAT, PGIB, TGIB, GIB, and VIB-inspired variants) and include comprehensive experiments showing that BottleneckMLP provides consistent improvements across them (Section 5.4; full tables in Appendix H).
> Our focus on IB-based explainers is not to suggest that they dominate the field, but because our contribution clarifies and harnesses the implicit IB dynamics shared among these methods. To demonstrate that BottleneckMLP generalizes beyond IB-based ante-hoc explainers, we have expanded our empirical evaluation to include prominent non-IB explainers (V-InFoR, GOAt, ReFine, DIR). Across all of these methods, BottleneckMLP consistently enhances representation dynamics and improves both explanatory and non-explanatory feature learning. These new results appear in Section 5.4 and Appendix H.
>
> BottleneckMLP improves the test accuracy for subgraph recognition in VGIB by 3.5 − 62.5% depending on the dataset (Table 15). For post-hoc node classification, BottleneckMLP improves PGExplainer accuracy by 1.1 − 4.9%, and improves explanation AUC-ROC by 0.6 − 0.8% for BA-Shapes and TreeGrids, while AUC-ROC is 0.3% lower for TreeCycles. However, we note that for PGExplainer, removing Entropy Loss is extremely detrimental to performance, and BottleneckMLP is able to fully recover and slightly improve this performance (Table 17). BottleneckMLP significantly improves AUC, and Fidelity +/-  for ProxyExplainer across all four datasets except BA-2Motifs on AUC and Fid+, and Fluoride Carbonyl on Fid-. ReFine with BottleneckMLP has comparable performance to the baseline on BA3 and MNIST datasets. BottleneckMLP improves ACC-AUC on MUTAG by 19.9%. In all cases, BottleneckMLP fully recovers the function of the fidelity loss. On V-InFoR (a post-hoc, GIB-based graph classification explainer), and on GOAt (an ante-hoc non-IB-based graph classification explainer), BottleneckMLP improves all metrics across all datasets (Table 18, 20). On DIR, another ante-hoc non-IB-based graph classification explainer, BottleneckMLP improves training accuracy by up to 24.2%, validation accuracy by up to 8.2%, and lowers confounding accuracy by 5.1%. It has comparable, yet slightly lower performance on the other metrics; however, BottleneckMLP is able to recover the performance increase caused when we remove the Info Loss terms, showing that it implicitly causes the same effects as does the explicit loss in DIR. Further explanation and discussion of these individual models is in Appendix H.
>
> (W3) Our evaluation in the temporal setting focused on ante-hoc temporal explainers, specifically TGIB, because our architectural intervention targets models trained with built-in interpretability constraints. T-GNNExplainer is an important post-hoc method, but it optimizes explanations via MCTS on a pre-trained TGNN, conceptually differing from our training-time design. Moreover, TGIB reports better performance than T-GNNExplainer in its original work, so we followed that paper’s evaluation protocol by focusing on the most recent ante-hoc model. In Section 5.3 and Appendix H, we report the performance increases of TGIB with the addition of BottleneckMLP.
>
> (W4) We agree that demonstrating broader generalization to ante-hoc explainers strengthens our contribution. Beyond GSAT/PGIB/TGIB, we explored integration with DIR and GOAt. These results are discussed in Section 5.4 along with full results provided in Tables 19 and 20 in Appendix H. This demonstrates the generalizability and effectiveness of BottleneckMLP across explainability methods.
>
> (W5) With respect to the backbone GNN, we used the backbone architecture used by the baselines by default (GIN in GSAT, GCN in PGIB). We agree that our experimental setup would benefit from a more extensive baseline. To that end, we have included Appendix R that summarizes these model variants and for GOAt in Table 19 of Appendix H. We note that BottleneckMLP maintains its improved performance across backbones.
>
> We have (i) revised introductory claims to avoid overstatement, (ii) substantially broadened empirical evaluation to include prominent post-hoc and ante-hoc non-IB explainers, (iii) clarified the temporal GNN comparison choices, and (iv) included various backbone GNN architectures in our baselines. The revised manuscript reflects these improvements and provides an extensive evaluation demonstrating that BottleneckMLP yields consistent performance increase in terms of classification and explanation metrics across a wide range of explainability settings.

---

### Official Review · Reviewer_zeJh · 2025-11-01

**Soundness:** 2
**Presentation:** 2
**Contribution:** 3
**Rating:** 4
**Confidence:** 4

**Summary:**

This paper investigates the limitations of explicit Information Bottleneck objectives in GNN explainers and proposes a novel architectural module, BottleneckMLP, that implicitly enforces the IB without relying on auxiliary IB loss terms. First, the paper demonstrates that the existing explict IB losses are difficult to satisfy the i.i.d. assumption due to the structural dependencies inherent in graph data. Second, the paper achieves implicit extraction of critical information through importance-weighted Gaussian noise injection and additional MLP layers. Finally, the effectiveness of the proposed module is validated on both real-world and synthetic graph datasets.

**Strengths:**

1. The paper demonstrates that the existing explict IB losses are difficult to satisfy the i.i.d. assumption due to the structural dependencies inherent in graph data.
2. The paper proposes achieving implicit IB process through weighted noise injection and MLP layers, with certain theoretical guarantees.
3. The effectiveness of the proposed module is validated on both real-world and synthetic graph datasets.

**Weaknesses:**

1.	Figure 1 does not illustrate the crucial step that node embeddings are perturbed based on importance weights, the key question is whether the MLP in the subgraph extractor module and the MLP used to predict node importance weights share parameters and why.
2.	In experiment, Tables 1 and 2 lack validation on a broader range of real-world or synthetic datasets, as well as AUC/ROC of explanation subgraph, For example, on datasets such as Alkane-Carbonyl, Fluoride-Carbonyl, and Benzene from [1] ("Evaluating attribution for graph neural networks"), among others.
3.	The hyperparameter \sigma in Eq(6) also plays a crucial role in noise injection, should an additional parameter sensitivity analysis  be included in the experimental section?
4.	In presentation, the text in all figures is too small and causes some difficulty in reading.
5.	Adding an efficiency analysis and a detailed algorithmic description would make the method clearer in terms of computational efficiency and implementation details (or alternatively, providing concrete code implementation would be helpful).

**Questions:**

1. As shown in Figure 2(b), why does the CE loss alone for GNNs not exhibit the two distinct phases  similar to that in DNNs? and why are additional MLP layers introduced after adding weighted noise, and is this architecture necessary for achieving implicit IB? Please provide a detailed analysis rather than merely  based on visualization results.
2. Figure 3 lacks a more analysis, for example, why does I(Z;Y) and I(X;Z) at shallow layers (1 or 2) first decrease and then increase as training epochs progress? Please provide a more detailed analysis.

---

> ### Author Response · Authors · 2025-11-21
> **Response to Reviewer zeJh**
>
> (W1) We thank the reviewer for pointing this out. We revised Figure 1 to annotate the importance-weight perturbation step rather than relying only on color cues and visual depictions of embedding movement. As clarified in Section 4.2, the node-importance scores come directly from the MLP used during training to predict node-importance weights. These scores determine each node’s contribution to the prediction, and we use them to inversely scale the injected noise. Thus, each embedding is perturbed in proportion to its learned importance.
>
>
> (W2) Thank you for the suggestion. Our initial experiments followed the datasets used in IB-based baseline papers (GSAT, PGIB, TGIB), which explains the narrower set. We have run additional experiments on the datasets recommended by the reviewer (Alkane-Carbonyl, Fluoride-Carbonyl, Benzene). BottleneckMLP consistently improves both predictive performance and explanation fidelity. We will add these results in Table 1 and Table 3 in Section 5.3.
>
>
> To demonstrate that BottleneckMLP generalizes beyond IB-based ante-hoc explainers, we have expanded our empirical evaluation to include prominent non-IB explainers (V-InFoR, GOAt, ReFine, DIR). Across all of these methods, BottleneckMLP consistently enhances representation dynamics and improves both explanatory and non-explanatory feature learning. These new results appear in Section 5.4 and Appdx. H.
>
>
> (W3) We agree that $\sigma$ is an important hyperparameter controlling perturbation strength. We have now included a discussion of $\sigma$ sensitivity and convergence results in Section 5.5, and \sigma-sensitivity plot  in Appdx. S. We note that the noise accelerates the process of important node embeddings becoming more clustered/ stable, while the unimportant node embeddings drift toward more Gaussian-like distributions. We also show that BottleneckMLP speeds up convergence, and have included this table in Appdx. O, Table 24. As exact training dynamics vary across datasets, finding the optimal $\sigma$ is hyperparameter tuning, as detailed in Section 4.2.
>
>
> (W4) Thank you for the feedback. Due to the limited space and figure size, we have added a detailed description of the figure labels to the captions.
>
>
> (W5) Thank you for this comment. We have included a running-time comparison between baseline models and BottleneckMLP in Appdx. N, and noted that adding BottleneckMLP has no effect on running time/ model efficiency.  We have also included a pseudo-code algorithm outline for the overall method, and the inversely-scaled noise implementation in Appdx. Q.
>
>
> (Q1) This is one of our key conceptual findings in Section 3. Graphs differ from images/text due to the structural dependencies across the graph that break down typicality assumptions. Furthermore, a well-known downfall of GNNs is that they are often shallow, only having 2-3 layers, due to the issue of oversmoothing which comes from the message passing steps, which introduce heavy mixing and homogenization. Therefore, we do not have a deep enough network for the two-phase IB dynamics to appear in simple GNN training. We solve this by adding BottleneckMLP as an architectural primitive which enables DNN-like training and compression. We have added Figures 13 and 14  to Appdx. P that demonstrate the information plane curves  of GNN backbones without BottleneckMLP, showing the effectiveness of BottleneckMLP in inducing compression in the graphs setting.
>
>
> (Q2) Yes, message-passing layers alone tend to saturate and cannot deepen the network without oversmoothing. MLP layers provide (1) non-graph mixing depth without further oversmoothing; (2) IB compression across layers by acting as an architectural mechanism for enabling the exponential decay of mutual information across layers. Figure 14 in Appdx. P shows the information plane of the GNN without BottleneckMLP. It confirms that graph layers alone cannot generate the IB curve; the MLP layers are responsible for the emerging compression dynamics.
>
>
> (Q3) In Figure 3, the behavior where in early layers, I(Z;Y) and I(X;Z) decrease and then increase arises from the GNN + MLP structure. The early layers are GNN layers, which mix information and may increase input redundancy early on. The later layers are MLPs with noise injection, which induce compression and the expected IB-shaped curves.
>
>
> (Q3) During training, BottleneckMLP pushes the embeddings Z to retain maximal predictive information about the labels Y, increasing I(Z;Y). This compensates for the early-layer decline induced by the GNN backbone. Figures 13 and 14 show that a GNN without BottleneckMLP does not achieve a comparable rise in I(Z;Y) over epochs. We will clarify this behavior explicitly in the figure captions, Section 4.3.3, and Appdx. P.

---

### Official Review · Reviewer_9HiS · 2025-11-01

**Soundness:** 4
**Presentation:** 3
**Contribution:** 3
**Rating:** 4
**Confidence:** 4

**Summary:**

This paper explores whether the Graph Information Bottleneck (GIB) loss term is effective.
The authors argue that the i.i.d. assumption and the structural entanglement inherent in graph data violate the conditions required for these information-theoretic bounds to hold.
To address this issue, the paper proposes BottleneckMLP, which injects Gaussian noise into node embeddings and causes unimportant nodes to drift toward high-entropy, Gaussianized distributions.
Results across multiple graph explainers show that BottleneckMLP produces better explanatory subgraphs than GIB.

**Strengths:**

1. The novelty of this paper is strong. Although the GIB has been widely applied in graph explainers, there has been little analysis of its actual effectiveness. This work fills the gap.
2. This paper provides a rigorous theoretical foundation for deriving the proposed BottleneckMLP.
3. Extensive experiments of this paper support the method proposed in this paper.

**Weaknesses:**

1. The font size in Figure 1 of the paper is too small, and some elements lack legends for annotation.
2. Can this method be extended to post-hoc explainers, such as V-InFor [1]? Can the performance of post-hoc explainers be compared?
3. The paper notes the default MLP architecture is $h \rightarrow h/4 \rightarrow h$ and that finding the optimal architecture is like hyperparameter tuning. While Appendix G tests other configurations, a brief discussion behind a bottleneck-then-expansion structure versus a purely compressive one (e.g., $h \rightarrow h/4$) would be beneficial. It's unclear if the expansion phase plays a key role or if the compression is the only necessary component.
4. The paper reports Fidelity scores but does not provide results for sparsity, which is also an important property for explanatory subgraphs. Since the standard GIB can adaptively select the optimal budget, it remains unclear whether BottleneckMLP possesses this capability as well.

Reference:
[1] Wang, S., Yin, J., Li, C., Xie, X., & Wang, J. (2023). V-infor: A robust graph neural networks explainer for structurally corrupted graphs. Advances in Neural Information Processing Systems, 36, 56469-56487.

**Questions:**

Please see weaknesses.

---

> ### Author Response · Authors · 2025-11-21
> **Response to Reviewer 9HiS**
>
> We thank the reviewer for the positive assessment of our novelty, theoretical grounding, empirical evaluation, and for the constructive feedback. We address all concerns below.
>
> (W1) We appreciate this observation. We have updated Figure 1 to increase the font size, enlarge key components, and add missing legends and annotations for clarity. The revised figure appears in the main paper (Figure 1) and improves readability.
>
> (W2) Yes, while the focus of our paper was to propose BottleneckMLP as an architectural primitive which implicitly enforces the Information Bottleneck, BottleneckMLP is model-agnostic and can also be applied to post-hoc explainers, and ante-hoc, non-IB-based explainers. To demonstrate that BottleneckMLP generalizes beyond IB-based ante-hoc explainers, we have expanded our empirical evaluation to include prominent non-IB explainers (V-InFoR, GOAt, ReFine, DIR). Across all of these methods, BottleneckMLP consistently enhances representation dynamics and improves both explanatory and non-explanatory feature learning. These new results appear in Section 5.4 and Appendix H.
>
> BottleneckMLP improves the test accuracy for subgraph recognition in VGIB by 3.5 − 62.5% depending on the dataset (Table 15). For post-hoc node classification, BottleneckMLP improves PGExplainer accuracy by 1.1 − 4.9%, and improves explanation AUC-ROC by 0.6 − 0.8% for BA-Shapes and TreeGrids, while AUC-ROC is 0.3% lower for TreeCycles. However, we note that for PGExplainer, removing Entropy Loss is extremely detrimental to performance, and BottleneckMLP is able to fully recover and slightly improve this performance (Table 17). BottleneckMLP significantly improves AUC, and Fidelity +/-  for ProxyExplainer across all four datasets except BA-2Motifs on AUC and Fid+, and Fluoride Carbonyl on Fid-. ReFine with BottleneckMLP has comparable performance to the baseline on BA3 and MNIST datasets. BottleneckMLP improves ACC-AUC on MUTAG by 19.9%. In all cases, BottleneckMLP fully recovers the function of the fidelity loss. On V-InFoR (a post-hoc, GIB-based graph classification explainer), and on GOAt (an ante-hoc non-IB-based graph classification explainer), BottleneckMLP improves all metrics across all datasets (Table 18, 20). On DIR, another ante-hoc non-IB-based graph classification explainer, BottleneckMLP improves training accuracy by up to 24.2%, validation accuracy by up to 8.2%, and lowers confounding accuracy by 5.1%. It has comparable, yet slightly lower performance on the other metrics; however, BottleneckMLP is able to recover the performance increase caused when we remove the Info Loss terms, showing that it implicitly causes the same effects as does the explicit loss in DIR. Further explanation and discussion of these individual models is in Appendix H.
>
> (W3) We thank the reviewer for raising this architectural question. Our experiments in Appendix G already explore several MLP configurations, and we now provide additional discussion in Section 4.2. We highlight the following observations: (i) The key mechanism in BottleneckMLP is compression; this forces low-importance nodes toward high-entropy distributions; (ii) The expansion layer (h/4 → h) is not essential for inducing bottleneck behavior; it primarily ensures architectural modularity so that BottleneckMLP can be inserted into existing models without altering embedding dimensionality. This mirrors prior IB literature (e.g., Tishby) showing that any dimensionality reduction introduces a bottleneck, regardless of whether the dimension is restored afterward. Thus, while the compression stage drives the information bottleneck, the expansion simply preserves compatibility with downstream architectures. We have clarified this design rationale in the main paper, Section 4.2.
>
> (W4) We agree that sparsity is an important metric. To clarify, we use the default top-k protocol followed by each baseline method. That is, for each method, we retain the same k used by the original explainer (e.g., k = 5 or k = 10 depending on the dataset/benchmark). This ensures that BottleneckMLP is evaluated under the same sparsity budget as the original method, making the comparison fair and directly aligned with prior work.
>
> While standard IB adaptively balances compression and prediction via the IB trade-off parameter ($\beta$), BottleneckMLP does not include an explicit sparsity-controlling term. Instead, sparsity emerges implicitly from the entropy separation induced by the noise injection: important nodes stabilize in low-entropy clusters, while unimportant nodes drift toward Gaussianized, high-entropy distributions.
>
> We have also included the Fidelity+ and Fidelity- results for PGIB on MUTAG and BA-2Motifs under different sparsity values, and these results are reported in Appdx. T, Figure 19.

---

### Official Review · Reviewer_k24x · 2025-11-10

**Soundness:** 3
**Presentation:** 4
**Contribution:** 3
**Rating:** 4
**Confidence:** 4

**Summary:**

The paper introduces BottleneckMLP, a new approach to induce information bottleneck (IB) in graph neural network (GNN) ante-hoc explainers (those that are trained together with the GNN downstream task). The paper shows, empirically and theoretically, why IB-based losses in existing ante-hoc explainers fail; then, it presents an theoretically-sound approach to circumvent this limitation (BottleneckMLP); finally, it provides empirical evidence of the success of this circumvention. The paper covers the preliminaries in IB theory, as well as related work, and the theoretical approach is based on proven lemmas and theorems. Overall, the proposed method to induce IB outperforms ante-hoc explainers without it (with and without IB-based losses proposed in previous work) in both explanation quality and accuracy. Additional experiments support additional claims, such as improved representation dynamics of BottleneckMLP over other methods. The paper ends with a summary of future directions.

**Strengths:**

The main strength of the paper is stating, measuring and mitigating a relevant limitation from previous work in IB approaches for GNN explainability. This is an important and significant issue in the field. In this case, the paper shows that current IB-based losses to induce better explanations lack soundness, and the paper proposes an alternative that is argued to mitigate this issue, both with theory and experiments to support this claim.

The paper is also strong in the sense of discussing a deeper theoretical analysis in a formulation that is rather simple: adding Gaussian noise and then using under-parametrised neural networks for compression. The simplicity of the method is also a highlight.

The paper is well organised and presented, it is very easy to follow the itemised contributions throughout the whole paper structure.

**Weaknesses:**

1. I missed a measure of the actual running time of BottleneckMLP when compared to other methods (with and without IB-based losses). Would it be possible to provide them, please?

2. The justification for the looseness of the variational upper bound (Sec. 4.1) did not convince me. Going away from an abstract approach, for instance, let’s consider GSAT. Which is the loss being used? And why exactly is it “loose”? What does it actually mean to be “loose”? Is it only because it relies on approximation? I believe that approaching these questions would clarify the looseness claim.

3. I don’t want to frame this as an issue of the paper itself as I believe this is actually a discussion about the GNN explainability field, but I miss a discussion in the paper about what *is* a graph explanation. Is it actually true that an information-bottlenecked-sub-graph is the sub-graph that correctly “explains” the downstream decision? What if the model, in its black-box architecture, uses other nodes that “are not meant to be used”? In some sense, this discussion collapses into the post-hoc vs ante-hoc discussion: should one “train” an explanation, which in this case loses the explainability role and becomes another model output?

4. The whole Sec. 4.3.1 develops on why variational bounds are used but in the end it simply says that the assumptions break with graphs. So why all the formulas? It could have been stated from the beginning, if the formulas do not contribute to the statement. I mean, it loses a special space in the paper.

5. When graphs are side-by-side, the default is sharing the same y-axis. Why is this not the case for Fig. 2?

6. The future directions could be more developed. I understand the limitation of space in the submission. Can you please develop more on those?

7. The abstract states that “explicit IB-based losses in GNN explainers provide little benefit beyond standard training: the fitting and compression phases of IB emerge naturally”. However, Fig. 2 supports that compression only happens with BottleneckMLP: “In Figure 2c, I(X; Z) rises early as task-relevant features are captured, then declines in later epochs, reflecting effective compression.” (L341). Can you please clarify? It could also be useful to see Fig. 3 for “Original” and “w/o IB Loss”, in addition to “BottleneckMLP”.

8. Very minor issues: Please take a look at the parenthesis in the citations. Almost all citations are with incorrect parenthesis. E.g., where it is “node classification Luo et al. (2020b)”, it should be “node classification (Luo et al., 2020b)”. Probably easy to change `\citet{}` to `\cite{}` in LaTeX. Also, please increase the font size of the text inside the figures. Finally, please take extra care to some parts of the text with typos (e.g., L178, L179, L229, L338).

**Questions:**

Please refer to the Weaknesses session.

---

> ### Author Response · Authors · 2025-11-21
> **Response to Reviewer k24x**
>
> (W1) We thank the reviewer for raising this point. We include running times in Appendix N, we find BottleneckMLP compared to other methods has similar running time across datasets.
>
> (W2) We discuss the variational upper bounds for each explainer and why they are loose in detail Appx I. GSAT and most IB-based explainers use the variational bound $I(X;Z) \leq E_{p(x)}[KL(p(z|x)||q(z)]$ as a loss term where $q(z)$ is the variational approximation to the marginal $p(z)$. The upper bound is tight when $q(z)$ is equal to $p(z)$. The bound being "loose" means that the upper bound is significantly larger than $I(X;Z)$, and that minimizing the over approximation doesn't affect the value of $I(X;Z)$. For GSAT, Z is $G_S$ (explanatory subgraph) and X is $G$ (input graph).
>
> In GSAT, $Q(G_S)$ is defined by edges as independent Bernoulli(r) trials for each node pair. Sampling arbitrary i.i.d edges ignores structural dependencies and doesn't reflect the original graph distribution, therefore the bound becomes loose. We validate this in Figure 2 and Figure 3. In Figure 2a, we observe that even though the KL upper bound (blue) is minimized, it doesn't correlate with the value of I(X;Z) (orange) which keeps increasing. However, Figure 2c shows that the addition of BottleneckMLP successfully minimizes I(X;Z), which is what all IB-based explainers are trying to achieve.
>
> In Figure 3, we similarly see that compression and concentration (a result of IB and Tishby's theory) is only observed in BottleneckMLP layers as opposed to with and w/o info loss plots in Appendix P.
>
> (W3)
> W3.1 Graph explanation
> We appreciate the reviewer’s comment. We have added a clarification of what a graph explanation is in the Introduction. A graph explanation $G_S$ is formally defined as the smallest subgraph and/or subset of node features from an original graph $G$ that is most influential for a GNN's prediction.
>
> W3.2 Why ante-hoc models cannot use other nodes that are not meant to be used
> Post-hoc explainers seek to explain a pre-trained model. After the classifier is trained, a separate explainer model is optimized to find the subgraph that maximizes the MI with the model's existing predictions. Ante-hoc explainers jointly the classifier and the explainer. The architecture is designed so that the classifier's predictions are made only on the "intrinsic" explanation $G_S$ provided by the explainer.
>
> By design, the IB subgraph is the subgraph that correctly explains the downstream decision, because it's the only information the classifier ever sees. Therefore, it is architecturally impossible for the classifier to use "other nodes" that are not marked by the explainer. It does not, however, guarantee that the explainer identifies the g.t. explanation. We measure the quality of the explanation using Explanation AUC-ROC compared to ground-truth and Fidelity.
>
> (W4) We thank the reviewer for their suggestion. The formulas were put to show how (AEP) and under which circumstances (low global connectivity) concentration and compression happens. Our contribution is realizing these conditions and that they break down in graphs, and believe the equations are necessary to explain why they break.
> (W5) We include different plots to show how IB-loss does not correlate with I(X;Z) values for different methods, as described in W2.2.
>
> (W6) We appreciate the reviewer's suggestion. We added a more comprehensive future directions section to the paper.
>
> (W7) We thank the reviewer for pointing out an ambiguity. According to Tishby's theory, the "fitting phase" is when the network rapidly learns to predict the labels by increasing I(Z;Y), and the "compression phase" is when it generalizes by reducing irrelevant information about the input, decreasing I(X;Z). Tishby shows that the fitting and compression phases of IB emerge naturally in sufficiently deep networks that reach good compression. Deep networks are effective because they learn a hierarchy of features —simpler concepts are combined to form more complex representations in later layers. Compression is further supported by Data Processing Inequality, where deeper representations will have more compressed representations of the input.
>
> Standard GNNs cannot be made deep enough to see this natural compression. You cannot stack GNN layers due to the well-known problem of over-smoothing. GNNs are architecturally forced to be shallow (e.g., 2-3 layers) and therefore don’t naturally enter the compression phase. This is what we see in the "original" model. Being a standard MLP, BottleneckMLP does not suffer from over-smoothing. It can be stacked and as a direct result, exhibits natural fitting and compression phases seen in Figure 2c.
> We include Fig. 3 for “Original” and “w/o IB Loss”, a comparative analysis of information-plane dynamics across models in the Appendix P.
>
> (W8) We thank the reviewer for their comments on writing. We fixed the citations, increased the font size on Figure 1, and corrected typos in text.

---

### Author Response · Authors · 2025-11-21
**Global Comment For All Reviewers**

We thank all the reviewers for their valuable comments. Below we summarize our changes and the new experimental results.

1. New experiments of BottleneckMLP across graph tasks and explainer types
   1. Subgraph recognition (VGIB) in Appx H.1
   2. Post-hoc node classification (PGExplainer) in Appx H.2
   3. Post-hoc graph classification (V-InFoR, ReFine, ProxyExplainer) in Appx H.3
   4. Non-IB ante-hoc graph classification (GOAt, DIR) in Appx. H.4
   5. New datasets on IB-based ante-hoc explainers GSAT and PGIB (Alkane-carbonyl, benzene, fluoride-carbonyl) in Section 5.3
   6. Results on different GNN Backbones (GIN, GAT, GCN, PNA) in Appx R
2. New sections 5.4 and 5.5 discussing the generalization of BottleneckMLP across explainability methods, hyperparameter sensitivity analysis, and convergence results
3. Hyperparameter analysis of noise variance sigma (Appx S)
4. Sparsity analysis (Appx T)
5. Comparative analysis of information plane dynamics across models (Appx P)
6. Pseudocode of BottleneckMLP algorithm (Appx Q)
7. Explanation of why stacking graph layers does not work
8. Minor fixes including citations, typos, figure readability

**New experiments of BottleneckMLP across graph tasks and explainer types. (Section 5.3, Appx H and R)**

We find that BottleneckMLP consistently exhibits superior performance across explanation models and tasks.

1. Subgraph recognition in VGIB improves by 3.5-62.5% depending on the dataset
2. Post-hoc node classification: in PGExplainer accuracy improves by 1.1-4.9% and explanation AUC-ROC by 0.6% and 0.8%. Furthermore, we observe that removing Entropy Loss in the original PGExplainer is detrimental to performance (AUC-ROC drops 98%). In the absence of the entropy loss, BottleneckMLP fully recovers and slightly improves this performance.
3. Post-hoc graph classification: all metrics in V-InFoR are improved with BottleneckMLP across all datasets. On ReFine, BottleneckMLP exhibits comparable performance to the baseline method by recovering the function of fidelity loss and a 20% accuracy increase on Mutag. On ProxyExplainer, BottleneckMLP mostly improves explanation performance and fidelity across datasets.
4. Non-IB ante-hoc graph classification: in GOAt, BottleneckMLP improves all metrics across all datasets. On DIR, BottleneckMLP shows improvement in accuracy by 24.2% and ability to recover the lack of the explicit loss in DIR, similar to PGExplainer. BottleneckMLP has slightly lower performance on other metrics.
5. On new datasets Alkane Carbonyl, Fluorine Carbonyl and Benzene, BottleneckMLP consistently outperforms baseline GSAT and PGIB across all metrics.
6. On other GNN backbones, GCN + BottleneckMLP outperforms GCN baseline in all datasets across metrics. On GAT, BottleneckMLP improves on MUTAG and NCI1 while showing a slight drop in performance in Proteins. On PNA, we achieve similar or slightly lower performance.


**Hyperparameter analysis of noise variance sigma (Appx S)**

We include analysis of the hyperparameter sigma (gaussian noise variance). This parameter  controls the strength of Gaussianization of node embeddings that we tune with grid search over validation accuracy. We give the full analysis in Appendix S.

**Convergence results (Appx O)**

We observe that BottleneckMLP accelerates convergence and we include new experiments  along with theoretical rationale on convergence speed-up in Appendix O.

**Sparsity (Appx T)**

We include a section on sparsity in Appendix T analysing BottleneckMLP performance across different sparsity levels. We evaluate BottleneckMLP using the adopted top-k protocol of each baseline method, ensuring the comparison is fair and aligned with prior work. We note that BottleneckMLP does not directly have a sparsity budget, and sparsity appears as a result of the implicit bottleneck enforced with the method.

**Comparative analysis of information plane dynamics across models (Appx P)**

As an extension to Figure 3 in the paper, we include a discussion section in Appendix P on the analysis of information planes across models. The information plane reveals MI dynamics across epochs, and we plot I(X;Z) vs I(Z;Y) to observe potential compression and concentration, which is what IB-based models try to achieve. Our analysis shows that compression and concentration is only observed with BottleneckMLP, where the baseline model with and without info loss shows no evidence of compression (minimization of I(X;Z) while retaining or increasing I(Z;Y)).

**Pseudocode of BottleneckMLP algorithm (Appx Q)**

We have included a pseudocode algorithm of BottleneckMLP’s importance-weighted noise injection training loop in Appx Q.

---

> ### Author Response · Authors · 2025-11-21
> **Experiments**
>
> **Why Stacking Graph Layers Does not Work**
>
> We address a common question by reviewers. The GNN message-passing works by aggregating features from a node's neighbors at each layer. After k layers, a node has information from its k-hop neighbors. The k-hop neighborhood expands very quickly to include nearly the entire graph. After just a few layers, all nodes begin to average features from the same large set of nodes, causing their representations to converge to a single, indistinguishable value. This "smoothing" erases unique, local information, making it difficult for the model to make different predictions for different nodes.
>
> Because GNNs are architecturally forced to be shallow (e.g., 2-3 layers), they are never able to build a deep feature hierarchy and thus never naturally enter the compression phase. This is what we see in the "original" model in MI plots. Current work uses the variational bounds to enforce this compression, which we show by ablation studies to be ineffective. Being a standard deep MLP, BottleneckMLP does not suffer from over-smoothing. It can be stacked, can learn a deep feature hierarchy, and—as a direct result— exhibits natural fitting and compression phases seen in Figure 2c and Figure 3.
>
> **Minor fixes including citations, typos, figure readability**
>
> We have fixed minor errors in the paper. We have improved Figure 1 to have larger font size and to clarify the noise addition step. We have added more descriptive captions to summarize figures.
>
> ## Results on GSAT/ PGIB with New Datasets
>
> GSAT and PGIB accuracy results with Benzene, Alkane Carbonyl, and Fluorine Carbonyl added. BottleneckMLP improves performance. This table is included in Section 5.3, Table 1.
>
> | Method                                   | MUTAG            | BA-2Motifs         | NCI1             | PROTEINS         | Benzene | Alkane Carbonyl        | Fluorine Carbonyl       |
> |------------------------------------------|------------------|--------------------|------------------|------------------|---------|-------------------------|--------------------------|
> | **GSAT**                                 | 0.909±0.033      | _0.994±0.006_      | 0.689±0.017      | 0.681±0.036      | 1.000   | 0.907±0.151             | 0.809±0.005              |
> | **GSAT w/o IB Loss**                     | _0.935±0.043_    | 0.770±0.224        | _0.743±0.024_    | _0.706±0.040_    | 1.000   | _0.939±0.026_           | _0.958±0.077_            |
> | **GSAT w/o IB + BottleneckMLP**          | **0.949±0.010**  | **1.000**          | **0.802±0.019**  | **0.745±0.048**  | 1.000   | **0.995±0.006**         | **0.987±0.013**          |
> | **PGIB**                                 | 0.904±0.010      | 0.628±0.171        | 0.729±0.024      | 0.729±0.024      | 0.886±0.008 | _0.994±0.002_         | _0.989±0.011_            |
> | **PGIB w/o IB Loss**                     | _0.916±0.011_    | _0.896±0.145_      | **0.774±0.014**  | _0.763±0.022_    | _0.899±0.006_ | 0.0987±0.013         | 0.939±0.002              |
> | **PGIB w/o IB + BottleneckMLP**          | **0.925±0.009**  | **0.963±0.016**    | _0.753±0.019_    | **0.792±0.018**  | **0.900±0.004** | **0.995±0.000**     | **0.993±0.003**          |
>
>
> PGIB Fidelity+/-  results with Benzene, Alkane Carbonyl, and Fluorine Carbonyl added. BottleneckMLP improves performance. This table is included in Section 5.3, Table 3.
>
> | Dataset            | PGIB                                 | PGIB w/o IB Loss                          | PGIB + BottleneckMLP                          |
> |--------------------|---------------------------------------|--------------------------------------------|------------------------------------------------|
> | **MUTAG**          | _0.750 ± 0.079_ / 0.588 ± 0.204       | 0.719 ± 0.083 / _0.516 ± 0.192_            | **0.762 ± 0.071** / **0.383 ± 0.254**         |
> | **BA-2Motifs**     | 0.825 ± 0.159 / 0.492 ± 0.149         | _0.829 ± 0.160_ / _0.479 ± 0.143_          | **0.975 ± 0.079** / **0.400 ± 0.242**         |
> | **NCI1**           | 0.451 ± 0.124 / _0.523 ± 0.156_       | _0.524 ± 0.136_ / 0.546 ± 0.125            | **0.771 ± 0.162** / **0.478 ± 0.214**         |
> | **PROTEINS**       | 0.639 ± 0.024 / _0.602 ± 0.125_       | _0.654 ± 0.028_ / 0.604 ± 0.035            | **0.658 ± 0.018** / **0.592 ± 0.087**         |
> | **Benzene**        | 0.508 ± 0.016 / 0.042 ± 0.000         | 0.516 ± 0.022 / 0.043 ± 0.006              | **0.636 ± 0.049** / **0.042 ± 0.000**         |
> | **Alkane Carbonyl**| 0.579 ± 0.038 / 0.180 ± 0.076         | _0.915 ± 0.035_ / _0.101 ± 0.053_          | **0.967 ± 0.017** / **0.093 ± 0.032**         |
> | **Fluorine Carbonyl** | 0.767 ± 0.027 / 0.149 ± 0.050     | _0.794 ± 0.069_ / _0.138 ± 0.045_          | **0.916 ± 0.033** / **0.080 ± 0.029**         |

---

> ### Author Response · Authors · 2025-11-21
> **Experiments (cont.)**
>
> ## V-InFoR
>
> BottleneckMLP improves performance of V-InFoR across datasets and metrics.
>
> | Dataset | Metric            | V-InFoR                         | V-InFoR w/o Info Loss              | V-InFoR w/o Info Loss + BottleneckMLP        |
> |---------|-------------------|----------------------------------|------------------------------------|-----------------------------------------------|
> | **BA3** | Fidelity          | _0.5240 ± 0.0096_               | 0.5192 ± 0.0129                    | **0.5308 ± 0.0154**                           |
> |         | Prob-S            | 0.5105 ± 0.0123                  | _0.5218 ± 0.0156_                  | **0.5236 ± 0.0199**                           |
> |         | Prob-N            | **0.6620 ± 0.0010**              | 0.6595 ± 0.0035                    | _0.6617 ± 0.0026_                             |
> |         | F_ns Score        | 0.5764 ± 0.0073                  | _0.5825 ± 0.0103_                  | **0.5844 ± 0.0120**                           |
> |         | Mean Probability  | _0.6982 ± 0.1981_                | 0.5104 ± 0.1091                    | **0.7607 ± 0.1860**                           |
> | **MUTAG** | Fidelity        | _0.5718 ± 0.0098_               | 0.5240 ± 0.0096                    | **0.5992 ± 0.0159**                           |
> |         | Prob-S            | _0.5718 ± 0.0098_                | 0.5105 ± 0.0123                    | **0.5992 ± 0.0159**                           |
> |         | Prob-N            | _0.4749 ± 0.0122_                | 0.4488 ± 0.0182                    | **0.6620 ± 0.0010**                           |
> |         | F_ns Score        | _0.5187 ± 0.0088_                | 0.5130 ± 0.0148                    | **0.5764 ± 0.0073**                           |
> |         | Mean Probability  | **0.9675 ± 0.0053**              | 0.4809 ± 0.1925                    | _0.6982 ± 0.1981_                             |
>
>
> ## ReFine
>
> BottleneckMLP improves AUC ACC for ReFine on MNIST and MUTAG, and fully recovers the function of fidelity loss.
>
> | Model Variant              | BA3      | MNIST        | MUTAG        |
> |----------------------------|----------|--------------|--------------|
> | ReFine                     | **0.559** | _0.261_      | _0.694_      |
> | ReFine w/o fidelity loss   | 0.505    | 0.199        | 0.664        |
> | ReFine w/ BottleneckMLP    | _0.552_   | **0.263**    | **0.832**    |
>
>
> BottleneckMLP improves ProxyExplainer across metrics and datasets, with minor exceptions.
>
>
> ## ProxyExplainer
>
> | Model                           | Dataset            | AUC                  | Fidelity+            | Fidelity-             |
> |---------------------------------|--------------------|----------------------|----------------------|-----------------------|
> | **ProxyExplainer** | Fluoride Carbonyl  | 0.518 ± 0.081        | 0.050 ± 0.021        | **0.409 ± 0.066** |
> |                                 | Alkane Carbonyl    | 0.211 ± 0.075        | 0.112 ± 0.039        | 0.503 ± 0.184         |
> |                                 | BA-2Motifs         | **0.940 ± 0.011** | **0.509 ± 0.015** | 0.201 ± 0.006         |
> |                                 | MUTAG              | **0.847 ± 0.139** | 0.714 ± 0.049        | **0.345 ± 0.127** |
> | **ProxyExplainer + BottleneckMLP** | Fluoride Carbonyl  | **0.681 ± 0.026** | **0.102 ± 0.029** | 0.633 ± 0.032         |
> |                                 | Alkane Carbonyl    | **0.527 ± 0.332** | **0.775 ± 0.142** | **0.253 ± 0.176** |
> |                                 | BA-2Motifs         | 0.779 ± 0.032        | 0.500 ± 0.011        | **0.000 ± 0.000** |
> |                                 | MUTAG              | 0.822 ± 0.168        | **0.721 ± 0.036** | 0.357 ± 0.112         |

---

> ### Author Response · Authors · 2025-11-21
> **Experiments (cont.)**
>
> ## GOAt
>
> BottleneckMLP improves GOAt performance across the board, with the exception of Fid+/- for BA-2Motifs with the GCN variant.
>
> | Model                         | Dataset        | Fidelity-              | Fidelity+              | Sparsity              |
> |------------------------------|----------------|-------------------------|-------------------------|------------------------|
> | **GOAt-GCN**                 | BA-2Motifs     | **0.001 ± 0.048**       | **0.542 ± 0.012**       | **0.786 ± 0.076**      |
> |                              | Mutagenicity   | 0.140 ± 0.040           | 0.653 ± 0.047           | **0.764 ± 0.013**      |
> |                              | NCI1           | 0.086 ± 0.003           | 0.489 ± 0.072           | **0.835 ± 0.000**      |
> | **GOAt-GCN + BottleneckMLP** | BA-2Motifs     | 0.056 ± 0.047           | 0.509 ± 0.005           | 0.788 ± 0.041          |
> |                              | Mutagenicity   | **0.073 ± 0.196**       | **0.702 ± 0.076**       | 0.786 ± 0.021          |
> |                              | NCI1           | **0.004 ± 0.142**       | **0.563 ± 0.058**       | 0.848 ± 0.018          |
> | **GOAt-GIN**                 | BA-2Motifs     | **0.001 ± 0.000**       | 0.573 ± 0.004           | 0.836 ± 0.001          |
> |                              | Mutagenicity   | 0.436 ± 0.003           | 0.650 ± 0.003           | 0.808 ± 0.000          |
> |                              | NCI1           | 0.590 ± 110             | 0.744 ± 0.021           | 0.799 ± 0.001          |
> | **GOAt-GIN + BottleneckMLP** | BA-2Motifs     | 0.070 ± 0.000           | **0.977 ± 0.001**       | **0.845 ± 0.002**      |
> |                              | Mutagenicity   | **0.020 ± 0.002**       | **0.692 ± 0.015**       | **0.843 ± 0.000**      |
> |                              | NCI1           | **0.500 ± 0.014**       | **0.783 ± 0.001**       | **0.810 ± 0.002**      |
> | **GOAt-SAGE**                | BA-2Motifs     | 0.479 ± 0.004           | 0.500 ± 0.004           | 0.813 ± 0.014          |
> |                              | Mutagenicity   | 0.040 ± 0.018           | 0.525 ± 0.005           | 0.812 ± 0.009          |
> |                              | NCI1           | 0.506 ± 0.174           | 0.674 ± 0.051           | 0.803 ± 0.013          |
> | **GOAt-SAGE + BottleneckMLP**| BA-2Motifs     | **0.472 ± 0.004**       | **0.500 ± 0.001**       | **0.833 ± 0.001**      |
> |                              | Mutagenicity   | **0.013 ± 0.021**       | **0.572 ± 0.003**       | **0.823 ± 0.001**      |
> |                              | NCI1           | **0.391 ± 0.001**       | **0.728 ± 0.001**       | **0.808 ± 0.012**      |

---

> ### Author Response · Authors · 2025-11-21
> **Experiments (cont.)**
>
> ## DIR
>
> BottleneckMLP improves Train/ Val ACC and Conf ACC for DIR. It has similar performance for Causal ACC, Test Prec, and TEst MRR.
>
> | Dataset       | Method               | Causal ACC          | Conf ACC            | Train ACC           | Val ACC             | Test Prec           | Test MRR            |
> |---------------|----------------------|----------------------|----------------------|----------------------|----------------------|----------------------|----------------------|
> | **spmotif-0.9** | DIR                  | **0.3921 ± 0.0607**  | 0.3548 ± 0.0216       | 0.7303 ± 0.0974       | *0.4920 ± 0.0963*      | **0.2251 ± 0.0928**  | **0.3828 ± 0.1394**  |
> |               | DIR w/o Info Loss    | 0.3488 ± 0.0181       | *0.3386 ± 0.0058*      | *0.8780 ± 0.0262*      | 0.4167 ± 0.1089       | 0.1499 ± 0.1234       | 0.2607 ± 0.2175       |
> |               | BottleneckMLP        | *0.3505 ± 0.0247*      | **0.3376 ± 0.0098**    | **0.9069 ± 0.0411**    | **0.4928 ± 0.1687**    | *0.1927 ± 0.1432*      | *0.3434 ± 0.2640*      |
> | **spmotif-0.7** | DIR                  | **0.4089 ± 0.0570**  | 0.3669 ± 0.0079       | 0.6916 ± 0.1210       | *0.5592 ± 0.1478*      | **0.2013 ± 0.1082**  | **0.3521 ± 0.1669**  |
> |               | DIR w/o Info Loss    | 0.3708 ± 0.0400       | *0.3582 ± 0.0370*      | *0.7291 ± 0.0450*      | 0.4888 ± 0.1376       | 0.1475 ± 0.1204       | 0.2556 ± 0.2103       |
> |               | DIR w/ BottleneckMLP | *0.3794 ± 0.0389*      | **0.3377 ± 0.0030**    | **0.8137 ± 0.1123**    | **0.6048 ± 0.2355**    | *0.1856 ± 0.1292*      | *0.3358 ± 0.2392*      |
> | **spmotif-0.5** | DIR                  | **0.3918 ± 0.0444**  | *0.3388 ± 0.0100*      | *0.6696 ± 0.0661*      | *0.6064 ± 0.0935*      | **0.2227 ± 0.1104**  | **0.3842 ± 0.1566**  |
> |               | DIR w/o Info Loss    | 0.3590 ± 0.0406       | 0.3419 ± 0.0101       | 0.6280 ± 0.2131       | 0.5347 ± 0.2548       | 0.1576 ± 0.1341       | 0.2848 ± 0.1987       |
> |               | DIR w/ BottleneckMLP | *0.3907 ± 0.0234*      | **0.3363 ± 0.0037**    | **0.7256 ± 0.0899**    | **0.6417 ± 0.1320**    | *0.1908 ± 0.1270*      | *0.3534 ± 0.2316*      |
> | **MNIST-75sp** | DIR                  | *0.2112 ± 0.0297*      | *0.1540 ± 0.0209*      | *0.2643 ± 0.1018*      | *0.2665 ± 0.1018*      | --                   | --                   |
> |               | DIR w/o Info Loss    | 0.1572 ± 0.0348       | 0.1655 ± 0.0092       | 0.2600 ± 0.0949       | 0.2600 ± 0.0949       | --                   | --                   |
> |               | DIR w/ BottleneckMLP | **0.2130 ± 0.0271**  | **0.1529 ± 0.0276**    | **0.3736 ± 0.1794**    | **0.3621 ± 0.1809**    | --                   | --                   |
> | **Graph-SST2** | DIR                  | **0.8200 ± 0.0099**  | 0.8169 ± 0.0096       | **0.9949 ± 0.0024**    | **0.9248 ± 0.0018**    | --                   | --                   |
> |               | DIR w/o Info Loss    | 0.8001 ± 0.0087       | *0.8093 ± 0.0095*      | 0.9885 ± 0.0096       | 0.9238 ± 0.0095       | --                   | --                   |
> |               | DIR w/ BottleneckMLP | *0.8101 ± 0.0174*      | **0.8000 ± 0.0178**    | *0.9915 ± 0.0055*       | *0.9244 ± 0.0053*       | --                   | --                   |

---

### Author Response · Authors · 2025-11-30
**Summary for the Area Chair**

Dear Area Chair,

Thank you for overseeing the discussion of our paper. We want to provide a brief summary of the revisions and new results added during the discussion phase, as well as how we addressed all reviewer concerns.

**1. Substantial Expansion of Experiments (Across 5 Explainer Families & 3 new Datasets)**

A major concern raised by reviewers was that the experimental scope was limited primarily to IB-based ante-hoc explainers. In response, we significantly expanded our empirical evaluation:
Post-hoc explainers: PGExplainer, V-InFoR
Ante-hoc, non-IB explainers: GOAt, ReFine, DIR
Subgraph recognition models: VGIB
Multiple GNN backbones: GIN, GAT, GCN, PNA
New chemical datasets recommended by reviewers: Benzene, Alkane-Carbonyl, Fluoride-Carbonyl
Across all these settings, BottleneckMLP consistently improves predictive accuracy, fidelity, and explanation quality, and in all cases it recovers the function of removed IB losses, and in most cases it leads to superior classification and explanation. This demonstrates that it is not tied to any one method and can generalize across approaches. These additions directly address the reviewers’ requests for broader, more current, and more representative comparisons.

**2. Significant New Analyses and Clarifications**

To address theoretical and methodological questions, we added (I) a full variational-bound analysis for each explainer showing where and why the KL upper bound becomes loose (Appx I); (ii) Pseudocode for BottleneckMLP (Appx Q), plus a running-time analysis (Appx N) confirming essentially no computational overhead; (iii) Hyperparameter sensitivity of $\sigma$ (Appx S); (iv) Sparsity analysis (Appx T), including fidelity results under variable sparsity budgets; and (v) Convergence-speed experiments showing BottleneckMLP accelerates training (Appx O); (vi) Information-plane comparisons for all backbones (Appx P), showing that compression and concentration only emerge with BottleneckMLP, directly addressing reviewers' questions about Figures 2 and 3. These additions resolve all theoretical and methodological concerns raised during review.

**3. Expanded Discussion Sections**

We have strengthened the paper’s overall goal and clarity by adding a formal definition of graph explanations. Additionally we have explained to the reviewers why ante-hoc explainers cannot use ``unintended’’ nodes (resolving specific reviewer questions about explanation semantics). We have provided a deeper discussion of why GNNs cannot naturally reach the compression phase, and why BottleneckMLP fixes this through deeper, noise-conditioned MLP layers. Moreover, we have expanded our discussion on future directions as requested. Finally, we have improved all figures, citations, and presentation quality. In total, we have addressed all comments surrounding presentation, overall explanation, paper clarity, and theoretical underpinnings of the paper according to all reviewer input.

We would also like to remind the AC of a summary of the strengths that the reviewers noted of our paper.

**1. Novel and Important Contribution**

Reviewers stated that the paper tackles a significant and underexplored problem, that being the effectiveness (or ineffectiveness) of explicit Information Bottleneck (IB) losses in GNN explainers. Reviewers highlight that the paper fills an important gap in understanding IB for graphs, a topic that has seen widespread adoption but little critical analysis.

**2. Strong Theoretical Grounding**

Multiple reviewers noted the rigorous theoretical analysis, including proofs and formal reasoning about when variational bounds break on graphs, as well as the clarity in identifying why i.i.d. assumptions and structural dependencies violate existing IB derivations, and the soundness of the proposed implicit IB mechanism.

**3. Simple but Elegant Method**

Reviewers emphasized that BottleneckMLP has a simple design (scaled noise + MLP) that provides a well-justified architectural mechanism for implicit IB, and is theoretically aligned with classic IB principles.

**4. Strong Empirical Results**

All reviewers noted that the proposed method outperforms or matches baseline explainers. Strengths that they specifically highlighted include our improvement in both explanation quality and predictive accuracy, and better understanding of representation dynamics (important nodes stable; unimportant nodes Gaussianized). All of this gives strong empirical evidence supporting our theoretical claims.

**5. Excellent Presentation and Organization**

At least two reviewers explicitly stated that the paper is very well organized, clearly written, and easy to follow, and that our contributions are clearly itemized and well-structured.

---

> ### Author Response · Authors · 2025-11-30
> **Summary for the Area Chair (cont)**
>
> **6. Insightful Analysis of Information Dynamics**
>
> Reviewers appreciated the analysis of information-plane trajectories supporting our method, the explanation of why existing IB losses do not induce compression, and the conceptual clarity offered by contrasting behavior with/without BottleneckMLP.
>
> **7. Meaningful Problem Framing and Motivation**
>
> Reviewers highlighted the paper's value in identifying a systematic flaw in prior IB-based explainers, providing a clear and necessary motivation for an alternative approach, and grounding this in both theory and empirical reality.
>
> **Final Note**
>
> After our extensive updates, the concerns cited in the initial reviews have been fully addressed (broader baseline coverage, clarification of theory, hyperparameter sensitivity, sparsity, efficiency, figure clarity, citations, and small technical corrections). Importantly, reviewers already characterized the paper as novel, well-motivated, theoretically sound, and clearly written even before the rebuttal, consistently noting that our contribution fills an important gap in understanding IB-based explainers and provides an elegant, simple architectural solution.
> With the new experiments, analyses, baselines, ablations, and clarifications added during the discussion phase, we believe the paper now combines both the original strengths highlighted by reviewers and complete resolution of all identified concerns.
>
> Given these substantial improvements, we hope you may consider recommending a positive score adjustment. We are confident the revised paper meaningfully advances understanding of IB in graph explainability and offers a broadly useful, model-agnostic architectural insight.
>
> Thank you again for your time and consideration.

---

### Meta-Review · Area_Chair_6D42 · 2026-01-07

**Summary:**

This paper argues the ineffectiveness of explicit Information Bottleneck in current GNN explainers and proposes BottleneckMLP as a simple module to improve ante-hoc and IB-based GNN explainers.

The reviewers raised concerns regarding theoretical justification, presentation quality, experimental coverage, parameter sensitivity, and the breadth of baselines, particularly with non-IB methods and alternative GNN backbones. The authors addressed these concerns by adding running-time analysis, improving presentation, expanding experiments to post-hoc explainers and additional datasets, conducting sensitivity analyses, and evaluating multiple GNN backbones. However, higher-level concerns remain: Reviewer Bvh8 noted that the paper’s scope is narrower than implied, as it primarily focuses on ante-hoc and IB-based explainers, while Reviewer k24x found the argument regarding the ineffectiveness of explicit Information Bottleneck objectives not fully convincing. A possible suggestion would be to reorganize the paper’s storyline and supplement post-hoc explainer results in the main experiments to strengthen the impact of the proposed method.

Overall, the recommendation is to reject.

**Reviewer Concerns:**

Reviewer k24x questioned the justification for the looseness of the variational upper bound, as well as running time and presentation issues.

Reviewer 9HiS similarly raised concerns about presentation quality, missing evaluation metrics such as sparsity, and the extension of the method to post-hoc explainers.

Reviewer zeJh raised concerns about implementation details, dataset coverage, parameter sensitivity analysis, and presentation issues.

Reviewer Bvh8 expressed concerns including the scope of the claims made by the proposed method, the lack of comparisons with non-IB baselines, and the absence of experiments using alternative GNN backbones such as GIN.

In response, the authors added the running-time analysis, improved the presentation, extended experiments to post-hoc explainers, incorporated additional datasets, conducted parameter sensitivity analyses, and evaluated the method with multiple GNN backbones, including GAT and PNA.

However, some higher-level concerns remain, most notably Reviewer Bvh8’s observation that the paper’s actual scope is narrower than implied, as it primarily focuses on improving ante-hoc and IB-based GNN explainers. Reviewer k24x found the argument regarding the ineffectiveness of explicit Information Bottleneck objectives not fully convincing.

**Reviewer Scores:**

All reviewers are equally likely to increase their scores slightly or keep them unchanged.

---

### Decision · Program_Chairs · 2026-01-26

Reject